# Efficient online learning with Kernels for adversarial large scale problems

**Rémi Jézéquel**      **Pierre Gaillard**      **Alessandro Rudi**

INRIA - Département d'Informatique de l'École Normale Supérieure
PSL Research University, Paris, France
`{remi.jezequel,pierre.gaillard,alessandro.rudi}@inria.fr`

## Abstract

We are interested in a framework of online learning with kernels for low-dimensional, but large-scale and potentially adversarial datasets. We study the computational and theoretical performance of online variations of kernel Ridge regression. Despite its simplicity, the algorithm we study is the first to achieve the optimal regret for a wide range of kernels with a per-round complexity of order $n^\alpha$ with $\alpha < 2$.

The algorithm we consider is based on approximating the kernel with the linear span of basis functions. Our contributions are twofold: 1) For the Gaussian kernel, we propose to build the basis beforehand (independently of the data) through Taylor expansion. For $d$-dimensional inputs, we provide a (close to) optimal regret of order $O((\log n)^{d+1})$ with per-round time complexity and space complexity $O((\log n)^{2d})$. This makes the algorithm a suitable choice as soon as $n \gg e^d$ which is likely to happen in a scenario with small dimensional and large-scale dataset; 2) For general kernels with low effective dimension, the basis functions are updated sequentially, adapting to the data, by sampling Nyström points. In this case, our algorithm improves the computational trade-off known for online kernel regression.

## 1 Introduction

Nowadays the volume and the velocity of data flows are deeply increasing. Consequently, many applications need to switch from batch to online procedures that can treat and adapt to data on the fly. Furthermore to take advantage of very large datasets, non-parametric methods are gaining increasing momentum in practice. Yet the latter often suffer from slow rates of convergence and bad computational complexities. At the same time, data is getting more complicated and simple stochastic assumptions, such as i.i.d. data, are often not satisfied. In this paper, we try to combine these different aspects due to large scale and arbitrary data. We build a non-parametric online procedure based on kernels, which is efficient for large data sets and achieves close to optimal theoretical guarantees.

Online learning is a subfield of machine learning where a learner sequentially interacts with an environment and tries to learn and adapt on the fly to the observed data as one goes along. We consider the following sequential setting. At each iteration $t \geq 1$, the learner receives some input $x_t \in \mathcal{X}$; makes a prediction $\widehat{y}_t \in \mathbb{R}$ and the environment reveals the output $y_t \in \mathbb{R}$. The inputs $x_t$ and the outputs $y_t$ are sequentially chosen by the environment and can be arbitrary. Learner's goal is to minimize his cumulative regret

$$R_n(f) := \sum_{t=1}^{n}(y_t - \widehat{y}_t)^2 - \sum_{t=1}^{n}\left(y_t - f(x_t)\right)^2 \tag{1}$$

uniformly over all functions $f$ in a space of functions $\mathcal{H}$. We will consider Reproducing Kernel Hilbert Space (RKHS) $\mathcal{H}$, [see next section or Aro50, for more details]. It is worth noting here that

all the properties of a RKHS are controlled by the associated *kernel function* $k : \mathcal{X} \times \mathcal{X} \to \mathbb{R}$, usually known in closed form, and that many function spaces of interest are (or are contained in) RKHS, e.g. when $\mathcal{X} \subseteq \mathbb{R}^d$: polynomials of arbitrary degree, band-limited functions, analytic functions with given decay at infinity, Sobolev spaces and many others [BTA11].

**Previous work**  Kernel regression in a statistical setting has been widely studied by the statistics community. Our setting of online kernel regression with adversarial data is more recent. Most of the existing work focuses on the linear setting (i.e., linear kernel). First work on online linear regression dates back to [Fos91]. [BKM+15] provided the minimax rates (together with an algorithm) and we refer the reader to references therein for a recent overview of the literature in the linear case. We only recall relevant work for this paper. [AW01, Vov01] designed the nonlinear Ridge forecaster (denoted *AWV*). In linear regression (linear kernel), it achieves the optimal regret of order $O(d \log n)$ uniformly over all $\ell_2$-bounded vectors. The latter can be extended to kernels (see Definition (3)) which we refer to as Kernel-AWV. With regularization parameter $\lambda > 0$, it obtains a regret upper-bounded for all $f \in \mathcal{H}$ as

$$R_n(f) \lesssim \lambda \|f\|^2 + B^2 d_{\text{eff}}(\lambda), \quad \text{where} \quad d_{\text{eff}}(\lambda) := \text{Tr}\big(K_{nn}\big(K_{nn} + \lambda I_n\big)^{-1}\big) \qquad (2)$$

is the effective dimension, where $K_{nn} := \big(k(x_i, x_j)\big)_{1 \le i,j \le n} \in \mathbb{R}^{n \times n}$ denotes the *kernel matrix* at time $n$. The above upper-bound on the regret is essentially optimal (see remark 2.1). Yet the per round complexity and the space complexity of Kernel-AWV are $O(n^2)$. In this paper, we aim at reducing this complexity while keeping optimal regret guarantees.

Though the literature on online contextual learning is vast, little considers non-parametric function classes. Related work includes [Vov06] that considers the Exponentially Weighted Average forecaster or [HM07] which considers bounded Lipschitz function set and Lipschitz loss functions, while here we focus on the square loss. Minimax rates for general function sets $\mathcal{H}$ are provided by [RST13]. RKHS spaces were first considered in [Vov05] though they only obtain $O(\sqrt{n})$ rates which are suboptimal for our problem. More recently, a regret bound of the form (2) was proved by [ZK10] for a clipped version of kernel Ridge regression and by [CLV17b] for a clipped version of Kernel Online Newton Step (KONS) for general exp-concave loss functions.

The computational complexity ($O(n^2)$ per round) of these algorithms is however prohibitive for large datasets. [CLV17b] and [CLV17a] provide approximations of KONS to get manageable complexities. However, these come with deteriorated regret guarantees. [CLV17b] improves the time and space complexities by a factor $\gamma \in (0, 1)$ enlarging the regret upper-bound by $1/\gamma$. [CLV17a] designs an efficient approximation of KONS based on Nyström approximation [SSL00, WS01] and restarts with per-round complexities $O\big(m^2\big)$ where $m$ is the number of Nyström points. Yet their regret bound suffers an additional multiplicative factor $m$ with respect to (2) because of the restarts. Furthermore, contrary to our results, the regret bounds of [CLV17b] and [CLV17a] are not with respect to all functions in $\mathcal{H}$ but only with functions $f \in \mathcal{H}$ such that $f(x_t) \le C$ for all $t \ge 1$ where $C$ is a parameter of their algorithm. Since $C$ comes has a multiplicative factor of their bounds, their results are sensitive to outliers that may lead to large $C$. Other relevant approximation schemes of Online Kernel Learning have been done by [LHW+16] and [ZL19]. The authors consider online gradient descent algorithms which they approximate using different approximation schemes (as Nyström and random features). However since they use general Lipschitz loss functions and consider $\ell_1$-bounded dual norm of functions $f$, their regret bounds of order $O(\sqrt{n})$ are hardly comparable to ours and seem suboptimal in $n$ in our restrictive setting with square loss and kernels with small effective dimension (such as Gaussian kernel).

**Contributions and outline of the paper**  The main contribution of the paper is to analyse a variant of Kernel-AWV that we call PKAWV (see Definition (4)). Despite its simplicity, it is to our knowledge the first algorithm for kernel online regression that recovers the optimal regret (see bound (2)) with an improved space and time complexity of order $\ll n^2$ per round. Table 1 summarizes the regret rates and complexities obtained by our algorithm and the ones of [CLV17b, CLV17a].

Our procedure consists simply in applying Kernel-AWV while, at time $t \ge 1$, approximating the RKHS $\mathcal{H}$ with a linear subspace $\tilde{\mathcal{H}}_t$ of smaller dimension. In Theorem 3, PKAWV suffers an additional approximation term with respect to the optimal bound of Kernel-AWV which can be made small enough by properly choosing $\tilde{\mathcal{H}}_t$. To achieve the optimal regret with low computational

| Kernel | Algorithm | Regret | Per-round complexity |
|---|---|---|---|
| Gaussian $d_{\text{eff}}(\lambda) \leq \left( \log \frac{n}{\lambda} \right)^d$ | PKAWV | $(\log n)^{d+1}$ | $(\log n)^{2d}$ |
| | Sketched-KONS [CLV17b] $(c > 0)$ | $c(\log n)^{d+1}$ | $\left( n/c \right)^2$ |
| | Pros-N-KONS [CLV17a] | $(\log n)^{2d+1}$ | $(\log n)^{2d}$ |
| General $d_{\text{eff}}(\lambda) \leq \left( \frac{n}{\lambda} \right)^\gamma$ $\gamma < \sqrt{2} - 1$ | PKAWV | $n^{\frac{\gamma}{\gamma+1}} \log n$ | $n^{\frac{4\gamma}{1-\gamma^2}}$ |
| | Sketched-KONS [CLV17b] $(c > 0)$ | $cn^{\frac{\gamma}{\gamma+1}} \log n$ | $\left( n/c \right)^2$ |
| | Pros-N-KONS [CLV17a] | $n^{\frac{4\gamma}{(1+\gamma)^2}} \log n$ | $n^{\frac{4\gamma(1-\gamma)}{(1+\gamma)^2}}$ |

Table 1: Order in $n$ of the best possible regret rates achievable by the algorithms and corresponding per-round time-complexity. Up to $\log n$, the rates obtained by PKAWV are optimal.

complexity, $\tilde{\mathcal{H}}_t$ needs to approximate $\mathcal{H}$ well and to be low dimensional with an easy-to-compute projection. We provide two relevant constructions for $\tilde{\mathcal{H}}_t$.

In section 3.1, we focus on the Gaussian kernel that we approximate by a finite set of basis functions. The functions are deterministic and chosen beforehand by the learner independently of the data. The number of functions included in the basis is a parameter to be optimized and fixes an approximation-computational trade-off. Theorem 4 shows that PKAWV satisfies (up to log) the optimal regret bounds (2) while enjoying a per-round space and time complexity of $O\left( \log^{2d} \left( \frac{n}{\lambda} \right) \right)$. For the Gaussian kernel, this corresponds to $O\left( d_{\text{eff}}(\lambda)^2 \right)$ which is known to be optimal even in the statistical setting with i.i.d. data.

In section 3.2, we consider data adaptive approximation spaces $\tilde{\mathcal{H}}_t$ based on Nyström approximation. At time $t \geq 1$, we approximate any kernel $\mathcal{H}$ by sampling a subset of the input vectors $\{x_1, \ldots, x_t\}$. If the kernel satisfies the capacity condition $d_{\text{eff}}(\lambda) \leq (n/\lambda)^\gamma$ for $\gamma \in (0, 1)$, the optimal regret is then of order $d_{\text{eff}}(\lambda) = O(n^{\gamma/(1+\gamma)})$ for well-tuned parameter $\lambda$. Our method then recovers the optimal regret with a computational complexity of $O\left( d_{\text{eff}}(\lambda)^{4/(1-\gamma)} \right)$. The latter is $o(n^2)$ (for well-tuned $\lambda$) as soon as $\gamma < \sqrt{2} - 1$. Furthermore, if the sequence of input vectors $x_t$ is given beforehand to the algorithm, the per-round complexity needed to reach the optimal regret is improved to $O(d_{\text{eff}}(\lambda)^4)$ and our algorithm can achieve it for all $\gamma \in (0, 1)$.

Finally, we perform in Section 4 several experiments based on real and simulated data to compare the performance (in regret and in time) of our methods with competitors.

**Notations** We recall here basic notations that we will use throughout the paper. Given a vector $v \in \mathbb{R}^d$, we write $v = (v^{(1)}, \ldots, v^{(d)})$. We denote by $\mathbb{N}_0 = \mathbb{N} \cup \{0\}$ the set of non-negative integers and for $p \in \mathbb{N}_0^d$, $|p| = p^{(1)} + \cdots + p^{(d)}$. By a sligh abuse of notation, we denote by $\| \cdot \|$ both the Euclidean norm and the norm for the Hilbert space $\mathcal{H}$. Write $v^\top w$, the dot product between $v, w \in \mathbb{R}^D$. The conjugate transpose for linear operator $Z$ on $\mathcal{H}$ will be denoted $Z^*$. The notation $\lesssim$ will refer to approximate inequalities up to logarithmic multiplicative factors. Finally, we will denote $a \vee b = \max(a, b)$ and $a \wedge b = \min(a, b)$, for $a, b \in \mathbb{R}$.

## 2 Background

**Kernels.** Let $k : \mathcal{X} \times \mathcal{X} \to \mathbb{R}$ be a positive definite kernel [Aro50] that we assume to be bounded (i.e., $\sup_{x \in \mathcal{X}} k(x, x) \leq \kappa^2$ for some $\kappa > 0$). The function $k$ is characterized by the existence of a *feature map* $\phi : \mathcal{X} \to \mathbb{R}^D$, with $D \in \mathbb{N} \cup \{\infty\}$[1] such that $k(x, x') = \phi(x)^\top \phi(x')$. Moreover the *reproducing kernel Hilbert space* (RKHS) associated to $k$ is characterized by $\mathcal{H} = \{f \mid f(x) = w^\top \phi(x), w \in \mathbb{R}^D, x \in \mathcal{X}\}$, with inner product $\langle f, g \rangle_{\mathcal{H}} := v^\top w$, for $f, g \in \mathcal{H}$ defined by $f(x) = v^\top \phi(x), g(x) = w^\top \phi(x)$ and $v, w \in \mathbb{R}^D$. For more details and different characterizations of $k, \mathcal{H}$, see [Aro50, BTA11]. It's worth noting that the knowledge of $\phi$ is not necessary when working with functions of the form $f = \sum_{i=1}^p \alpha_i \phi(x_i)$, with $\alpha_i \in \mathbb{R}$, $x_i \in \mathcal{X}$ and finite $p \in \mathbb{N}$, indeed $f(x) = \sum_{i=1}^p \alpha_i \phi(x_i)^\top \phi(x) = \sum_{i=1}^p \alpha_i k(x_i, x)$, and moreover $\|f\|_{\mathcal{H}}^2 = \alpha^\top K_{pp} \alpha$, with $K_{pp}$ the kernel matrix associated to the set of points $x_1, \ldots, x_p$.

**Kernel-AWV.** The Azoury-Warmuth-Vovk forecaster (denoted *AWV*) on the space of linear functions on $\mathcal{X} = \mathbb{R}^d$ has been introduced and analyzed in [AW01, Vov01]. We consider here a straightforward generalization to kernels (denoted *Kernel-AWV*) of the nonlinear Ridge forecaster (*AWV*) introduced by [AW01, Vov01] on the space of linear functions on $\mathcal{X} = \mathbb{R}^d$. At iteration $t \geq 1$, Kernel-AWV predicts $\widehat{y}_t = \widehat{f}_t(x_t)$, where

$$\widehat{f}_t \in \underset{f \in \mathcal{H}}{\operatorname{argmin}} \left\{ \sum_{s=1}^{t-1} \big(y_s - f(x_s)\big)^2 + \lambda \|f\|^2 + f(x_t)^2 \right\}. \tag{3}$$

A variant of this algorithm, more used in the context of data independently sampled from distribution, is known as *kernel Ridge regression*. It corresponds to solving the problem above, without the last penalization term $f(x_t)^2$.

**Optimal regret for Kernel-AWV.** In the next proposition, we state a preliminary result which proves that Kernel-AWV achieves a regret depending on the eigenvalues of the kernel matrix.

**Proposition 1.** *Let $\lambda, B > 0$. For any RKHS $\mathcal{H}$, for all $n \geq 1$, for all inputs $x_1, \ldots, x_n \in \mathcal{X}$ and all $y_1, \ldots, y_n \in [-B, B]$, the regret of Kernel-AWV is upper-bounded for all $f \in \mathcal{H}$ as*

$$R_n(f) \leq \lambda \|f\|^2 + B^2 \sum_{k=1}^{n} \log \left( 1 + \frac{\lambda_k(K_{nn})}{\lambda} \right),$$

*where $\lambda_k(K_{nn})$ denotes the $k$-th largest eigenvalue of $K_{nn}$.*

The proof is a direct consequence of the known regret bound of *AWV* in the finite dimensional linear regression setting—see Theorem 11.8 of [CBL06] or Theorem 2 of [GGHS18]. For completeness, we reproduce the analysis for infinite dimensional space (RKHS) in Appendix C.1. In online linear regression in dimension $d$, the above result implies the optimal rate of convergence $dB^2 \log(n) + O(1)$ (see [GGHS18] and [Vov01]). As shown by the following proposition, Proposition 1 yields optimal regret (up to log) of the form (2) for online kernel regression.

**Proposition 2.** *For all $n \geq 1$, $\lambda > 0$ and all input sequences $x_1, \ldots, x_n \in \mathcal{X}$,*

$$\sum_{k=1}^{n} \log \left( 1 + \frac{\lambda_k(K_n)}{\lambda} \right) \leq \log \left( e + \frac{en\kappa^2}{\lambda} \right) d_{\textit{eff}}(\lambda).$$

Combined with Proposition 1, this entails that Kernel-AWV satisfies (up to the logarithmic factor) the optimal regret bound (2). As discussed in the introduction, such an upper-bound on the regret is not new and was already proved by [ZK10] or by [CLV17b] for other algorithms. An advantage of Kernel-AWV is that it does not require any clipping and thus the beforehand knowledge of $B > 0$ to obtained Proposition 1. Furthermore, we slightly improve the constants in the above proposition.

*Remark* 2.1 (Optimal regret under the capacity condition). Assuming the capacity condition ($d_{\text{eff}}(\lambda) \leq (n/\lambda)^\gamma$ for $0 \leq \gamma \leq 1$), the rate of the regret bound (2) can be made explicit. As we show now, this matches existing minimax lower rates in the stochastic setting. Under the capacity condition, optimizing $\lambda \simeq n^{\gamma/(1+\gamma)}$ to minimize the r.h.s. of (2), the regret bound is then of order $R_n(f) \leq O(n^{\gamma/(1+\gamma)})$ (up to logs). If the data $(x_1, y_1), \ldots, (x_n, y_n)$ is i.i.d. according to some distribution $\rho$ over $\mathcal{X} \times \mathbb{R}$, we can apply a standard online to batch conversion (see [CBCG04]). The estimator $\bar{f}_n = \frac{1}{n} \sum_{t=1}^{n} f_t$ satisfies for any $f \in \mathcal{H}$ the upper-bound on its excess risk

$$\mathcal{E}(\bar{f}_n) - \mathcal{E}(f) \leq \mathbb{E}\left[ \frac{R_n(f)}{n} \right] \leq O(n^{-\frac{1}{1+\gamma}}),$$

where $\mathcal{E}(f) := \mathbb{E}_{(X,Y) \sim \rho}\big[(f(X) - Y)^2\big]$. This corresponds to the known minimax lower rate in this stochastic setting as shown by Theorem 2 (applied with $c = 1$ and $b = 1/\gamma$) of [CDV07].

It is worth pointing out that in the worst case $d_{\text{eff}}(\lambda) \leq \kappa^2 n/\lambda$ for any bounded kernel. In particular, optimizing the bound yields $\lambda = O(\sqrt{n \log n})$ and a regret bound of order $O(\sqrt{n \log n})$. In the special case of the Gaussian kernel (which we consider in Section 3.1), the latter can be improved to $d_{\text{eff}}(\lambda) \lesssim \big(\log(n/\lambda)\big)^d$ (see [ABRW18]) which entails $R_n(f) \leq O\big((\log n)^{d+1}\big)$ for well tuned value of $\lambda$.

# 3 Online Kernel Regression with projections

In the previous section we have seen that Kernel-AWV achieves optimal regret. Yet, it has computational requirements that are $O(n^3)$ in time and $O(n^2)$ in space, for $n$ steps of the algorithm, making it unfeasible in the context of large scale datasets, i.e. $n \gg 10^5$. In this paper, we consider and analyze a simple variation of Kernel-AWV denoted PKAWV. At time $t \geq 1$, for a regularization parameter $\lambda > 0$ and a linear subspace $\tilde{\mathcal{H}}_t$ of $\mathcal{H}$ the algorithm predicts $\hat{y}_t = \hat{f}_t(x_t)$, where

$$\hat{f}_t = \operatorname*{argmin}_{f \in \tilde{\mathcal{H}}_t} \left\{ \sum_{s=1}^{t-1} (y_s - f(x_s))^2 + \lambda \|f\|^2 + f(x_t)^2 \right\}. \tag{4}$$

In the next subsections, we explicit relevant approximations $\tilde{\mathcal{H}}_t$ (typically the span of a small number of basis functions) of $\mathcal{H}$ that trade-off good approximation with a low computational cost. Appendix H details how (4) can be efficiently implemented in these cases.

The result below bounds the regret of the PKAWV for any function $f \in \mathcal{H}$ and holds for any bounded kernel and any explicit subspace $\tilde{\mathcal{H}}$ associated with projection $P$. The cost of the approximation of $\mathcal{H}$ by $\tilde{\mathcal{H}}$ is measured by the important quantity $\mu := \left\| (I - P)C_n^{1/2} \right\|^2$, where $C_n$ is the covariance operator.

**Theorem 3.** *Let $\tilde{\mathcal{H}}$ be a linear subspace of $\mathcal{H}$ and $P$ the Euclidean projection onto $\tilde{\mathcal{H}}$. When PKAWV is run with $\lambda > 0$ and fixed subspaces $\tilde{\mathcal{H}}_t = \tilde{\mathcal{H}}$, then for all $f \in \mathcal{H}$*

$$R_n(f) \leq \lambda \|f\|^2 + B^2 \sum_{j=1}^{n} \log \left( 1 + \frac{\lambda_j(K_{nn})}{\lambda} \right) + (\mu + \lambda) \frac{n \mu B^2}{\lambda^2}, \tag{5}$$

*for any sequence $(x_1, y_1), \ldots, (x_n, y_n) \in \mathcal{X} \times [-B, B]$ where $\mu := \left\| (I - P)C_n^{1/2} \right\|^2$ and $C_n := \sum_{t=1}^{n} \phi(x_t) \otimes \phi(x_t)$.*

The proof of Thm. 3 is deferred to Appendix D.1 and is the consequence of a more general Thm. 9.

## 3.1 Learning with Taylor expansions and Gaussian kernel for very large data set

In this section we focus on non-parametric regression with the widely used *Gaussian kernel* defined by $k(x, x') = \exp(-\|x - x'\|^2/(2\sigma^2))$ for $x, x' \in \mathcal{X}$ and $\sigma > 0$ and the associated RKHS $\mathcal{H}$.

Using the results of the previous section with a fixed linear subspace $\tilde{\mathcal{H}}$ which is the span of a basis of $O(\operatorname{polylog}(n/\lambda))$ functions, we prove that PKAWV achieves optimal regret. This leads to a computational complexity that is only $O(n \operatorname{polylog}(n/\lambda))$ for optimal regret. We need a basis that (1) approximates very well the Gaussian kernel and at the same time (2) whose projection is easy to compute. We consider the following basis of functions, for $k \in \mathbb{N}_0^d$,

$$g_k(x) = \prod_{i=1}^{d} \psi_{k_i}(x^{(i)}), \quad \text{where} \quad \psi_t(x) = \frac{x^t}{\sigma^t \sqrt{t!}} e^{-\frac{x^2}{2\sigma^2}}. \tag{6}$$

For one dimensional data, this corresponds to Taylor expansion of the Gaussian kernel. Our theorem below states that PKAWV (see (4)) using for all iterations $t \geq 1$

$$\tilde{\mathcal{H}}_t = \operatorname{Span}(G_M) \qquad \text{with} \quad G_M = \{g_k \mid |k| \leq M, k \in \mathbb{N}_0^d\}$$

where $|k| := k_1 + \cdots + k_d$, for $k \in \mathbb{N}_0^d$, gets optimal regret while enjoying low complexity. The size of the basis $M$ controls the trade-off between approximating well the Gaussian kernel (to incur low regret) and large computational cost. Theorem 4 optimizes $M$ so that the approximation term of Theorem 3 (due to kernel approximation) is of the same order than the optimal regret.

**Theorem 4.** *Let $\lambda > 0, n \in \mathbb{N}$ and let $R, B > 0$. Assume that $\|x_t\| \leq R$ and $|y_t| \leq B$. When $M = \left\lceil \frac{8R^2}{\sigma^2} \vee 2 \log \frac{n}{\lambda \wedge 1} \right\rceil$, then running PKAWV using $G_M$ as set of functions achieves a regret bounded by*

$$\forall f \in \mathcal{H}, R_n(f) \leq \lambda \|f\|^2 + \frac{3B^2}{2} \sum_{j=1}^{n} \log \left( 1 + \frac{\lambda_j(K_{nn})}{\lambda} \right).$$

*Moreover, its per iteration computational cost is $O\left( \left( 3 + \frac{1}{d} \log \frac{n}{\lambda \wedge 1} \right)^{2d} \right)$ in space and time.*

Therefore PKAWV achieves a regret-bound only deteriorated by a multiplicative factor of $3/2$ with respect to the bound obtained by Kernel-AWV (see Prop. 1). From Prop. 2 this also yields (up to log) the optimal bound (2).

In particular, it is known [ABRW18] for the Gaussian kernel that

$$d_{\text{eff}}(\lambda) \leq 3\Big(6 + \frac{41}{d}\frac{R^2}{2\sigma^2} + \frac{3}{d}\log\frac{n}{\lambda}\Big)^d = O\Big(\Big(\log\frac{n}{\lambda}\Big)^d\Big).$$

The upper-bound is matching even in the i.i.d. setting for nontrivial distributions. In this case, we have $|G_M| \lesssim d_{\text{eff}}(\lambda)$. The per-round space and time complexities are thus $O\big(d_{\text{eff}}(\lambda)^2\big)$. Though our method is quite simple (since it uses fixed explicit embedding) it is able to recover results -in terms of computational time and bounds in the adversarial setting- that are similar to results obtained in the more restrictive i.i.d. setting obtained via much more sophisticated methods, like learning with (1) Nyström with importance sampling via leverage scores [RCR15], (2) reweighted random features [Bac17, RR17], (3) volume sampling [DWH18]. By choosing $\lambda = (B/\|f\|)^2$, to minimize the r.h.s. of the regret bound of the theorem, we get

$$R_n(f) \lesssim \Big(\log\frac{n\|f\|_{\mathcal{H}}^2}{B^2}\Big)^{d+1} B^2 + B^2. \tag{7}$$

Note that the optimal $\lambda$ does not depend on $n$ and can be optimized in practice through standard online calibration methods. For instance, one can run in parallel subroutines of the algorithm, each using a different value of $\lambda$ in the finite grid $\Lambda := \{n2^k, k = -n^{\frac{1}{d+1}}, \ldots, 0\}$. The subroutines can then be sequentially combined with an expert advice algorithm such as the Exponentially Weighted Average forecaster [CBL06] at an additional negligible cost of order $O(B^2 \log |\Lambda|)$ in the regret (using the fact that the squared loss is exp-concave on $[0, B]$). Similarly, though we use a fixed number of features $M$ in the experiments, the latter could be increased slowly over time thanks to online calibration techniques.

## 3.2 Nyström projection

The previous two subsections considered a deterministic function basis (independent of the data) to approximate specific RKHS. Here, we analyse Nyström projections [RCR15] that are data dependent and work for any RKHS. It consists in sequentially updating a dictionary $\mathcal{I}_t \subset \{x_1, \ldots, x_t\}$ and using

$$\tilde{\mathcal{H}}_t = \text{Span}\Big\{\phi(x), \ x \in \mathcal{I}_t\Big\}. \tag{8}$$

If the points included into $\mathcal{I}_t$ are well-chosen, the latter may approximate well the solution of (3) which belongs to the linear span of $\{\phi(x_1), \ldots, \phi(x_t)\}$. The inputs $x_t$ might be included in the dictionary independently and uniformly at random. Here, we build the dictionary by following the KORS algorithm of [CLV17a] which is based on approximate leverage scores. At time $t \geq 1$, it evaluates the importance of including $x_t$ to obtain an accurate projection $P_t$ by computing its leverage score. Then, it decides to add it or not, by drawing a Bernoulli random variable. The points are never dropped from the dictionary so that $\mathcal{I}_1 \subset \mathcal{I}_2 \subset \cdots \mathcal{I}_n$. With their notations, choosing $\varepsilon = 1/2$ and remarking that $\|\Phi_t^T(I - P_t)\Phi_t\| = \|(I - P_t)C_t^{1/2}\|^2$, their Proposition 1 can be rewritten as follows.

**Proposition 5.** *[CLV17a, Prop. 1] Let $\delta > 0$, $n \geq 1$, $\mu > 0$. Then, the sequence of dictionaries $\mathcal{I}_1 \subset \mathcal{I}_2 \subset \cdots \subset \mathcal{I}_n$ learned by KORS with parameters $\mu$ and $\beta = 12\log(n/\delta)$ satisfies w.p. $1 - \delta$,*

$$\forall t \geq 1, \qquad \big\|(I - P_t)C_t^{1/2}\big\|^2 \leq \mu \qquad \text{and} \qquad |\mathcal{I}_t| \leq 9d_{\text{eff}}(\mu)\log\big(2n/\delta\big)^2.$$

*Furthermore, the algorithm runs in $O\big(d_{\text{eff}}(\mu)^2 \log^4(n)\big)$ space and $O\big(d_{\text{eff}}(\mu)^2\big)$ time per iteration.*

Using this approximation result together with Thm. 9 (which is a more general version of Thm. 3), we can bound the regret of PKAWV with KORS. The proof is postponed to Appendix E.1.

**Theorem 6.** *Let $n \geq 1$, $\delta > 0$ and $\lambda \geq \mu > 0$. Assume that the dictionaries $(\mathcal{I}_t)_{t \geq 1}$ are built according to Proposition 5. Then, probability at least $1 - \delta$, PKAWV with the subspaces $\tilde{\mathcal{H}}_t$ defined in (8) satisfies the regret upper-bound*

$$R_n \leq \lambda \|f\|^2 + B^2 d_{\text{eff}}(\lambda) \log\big(e + en\kappa^2/\lambda\big) + 2B^2(|\mathcal{I}_n| + 1)\frac{n\mu}{\lambda},$$

*and the algorithm runs in $O(d_{\text{eff}}(\mu)^2)$ space $O(d_{\text{eff}}(\mu)^2)$ time per iteration.*

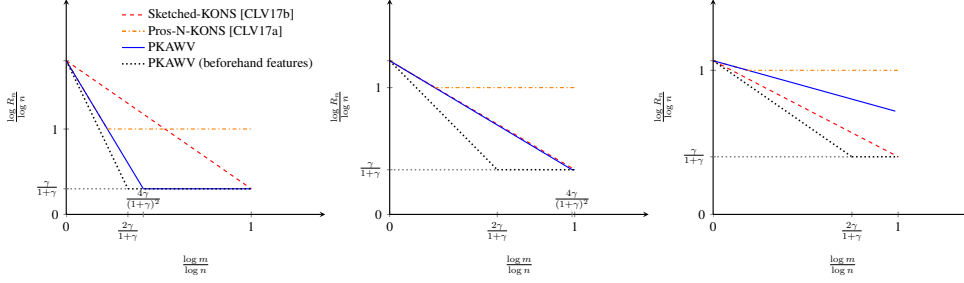

Figure 1: Comparison of the theoretical regret rate $\log R_n/\log n$ according to the size of the dictionary $\log m/\log n$ considered by PKAWV, Sketched-KONS and Pros-N-KONS for optimized parameters when $d_{\text{eff}}(\lambda) \le (n/\lambda)^\gamma$ with $\gamma = 0.2, \sqrt{2}-1, 0.6$ (from left to right). The value $\gamma/(1+\gamma)$ corresponds to the optimal rate.

The last term of the regret upper-bound above corresponds to the approximation cost of using the approximation (8) in PKAWV. This cost is controlled by the parameter $\mu > 0$ which trades-off between having a small approximation error (small $\mu$) and a small dictionary of size $|\mathcal{I}_n| \approx d_{\text{eff}}(\mu)$ (large $\mu$) and thus a small computational complexity. For the Gaussian Kernel, using that $d_{\text{eff}}(\lambda) \le O\big(\log(n/\lambda)^d\big)$, the above theorem yields for the choice $\lambda = 1$ and $\mu = n^{-2}$ a regret bound of order $R_n \le O\big((\log n)^{d+1}\big)$ with a per-round time and space complexity of order $O(|\mathcal{I}_n|^2) = O\big((\log n)^{2d+4}\big)$. We recover a similar result to the one obtained in Section 3.1.

**Explicit rates under the capacity condition**   Assuming the capacity condition $d_{\text{eff}}(\lambda') \le \big(n/\lambda'\big)^\gamma$ for $0 \le \gamma \le 1$ and $\lambda' > 0$, which is a classical assumption made on kernels [RCR15], the following corollary provides explicit rates for the regret according to the size of the dictionary $m \approx |\mathcal{I}_n|$.

**Corollary 7.** *Let $n \ge 1$ and $m \ge 1$. Assume that $d_{\text{eff}}(\lambda') \le (n/\lambda')^\gamma$ for all $\lambda' > 0$. Then, under the assumptions of Theorem 6, PKAWV with $\mu = nm^{-1/\gamma}$ has a dictionary of size $|\mathcal{I}_n| \lesssim m$ and a regret upper-bounded with high-probability as*

$$R_n \lesssim \begin{cases} n^{\frac{\gamma}{1+\gamma}} & \text{if } m \ge n^{\frac{2\gamma}{1-\gamma^2}} & \text{for } \lambda = n^{\frac{\gamma}{1+\gamma}} \\ nm^{\frac{1}{2}-\frac{1}{2\gamma}} & \text{otherwise} & \text{for } \lambda = nm^{\frac{1}{2}-\frac{1}{2\gamma}} \end{cases} .$$

*The per-round space and time complexity of the algorithm is $O(m^2)$ per iteration.*

The rate of order $n^{\frac{\gamma}{1+\gamma}}$ is optimal in this case (it corresponds to optimizing (2) in $\lambda$). If the dictionary is large enough $m \ge n^{2\gamma/(1-\gamma^2)}$, the approximation term is negligible and the algorithm recovers the optimal rate. This is possible for a small dictionary $m = o(n)$ whenever $2\gamma/(1-\gamma^2) < 1$, which corresponds to $\gamma < \sqrt{2}-1$. The rates obtained in Corollary 7 can be compared to the one obtained by Sketched-KONS of [CLV17b] and Pros-N-KONS of [CLV17a] which also provide a similar trade-off between the dictionary size $m$ and a regret bound. The forms of the regret bounds in $m, \mu, \lambda$ of the algorithms can be summarized as follows

$$R_n \lesssim \begin{cases} \lambda + d_{\text{eff}}(\lambda) + \frac{nm\mu}{\lambda} & \text{for PKAWV with KORS} \\ \lambda + \frac{n}{m}d_{\text{eff}}(\lambda) & \text{for Sketched-KONS} \\ m(\lambda + d_{\text{eff}}(\lambda)) + \frac{n\mu}{\lambda} & \text{for Pros-N-KONS} \end{cases} . \tag{9}$$

When $d_{\text{eff}}(\lambda) \le (n/\lambda)^\gamma$, optimizing these bounds in $\lambda$, PKAWV performs better than Sketched-KONS as soon as $\gamma \le 1/2$ and the latter cannot obtain the optimal rate $\lambda + d_{\text{eff}}(\lambda) = n^{\frac{\gamma}{1+\gamma}}$ if $m = o(n)$. Furthermore, because of the multiplicative factor $m$, Pros-N-KONS can't either reached the optimal rate even for $m = n$. Figure 1 plots the rate in $n$ of the regret of these algorithms when enlarging the size $m$ of the dictionary. We can see that for $\gamma = 1/4$, PKAWV is the only algorithm that achieves the optimal rate $n^{\gamma/(1+\gamma)}$ with $m = o(n)$ features. The rate of Pros-N-KONS cannot beat $4\gamma/(1+\gamma)^2$ and stops improving even when the size of the dictionary increases. This is because Pros-N-KONS is restarted whenever a point is added in the dictionary which is too costly for large dictionaries. It is worth pointing out that these rates are for a well-tuned value of $\lambda$. However, such an optimization can be performed at a small cost using expert advice algorithm on a finite grid of $\lambda$.

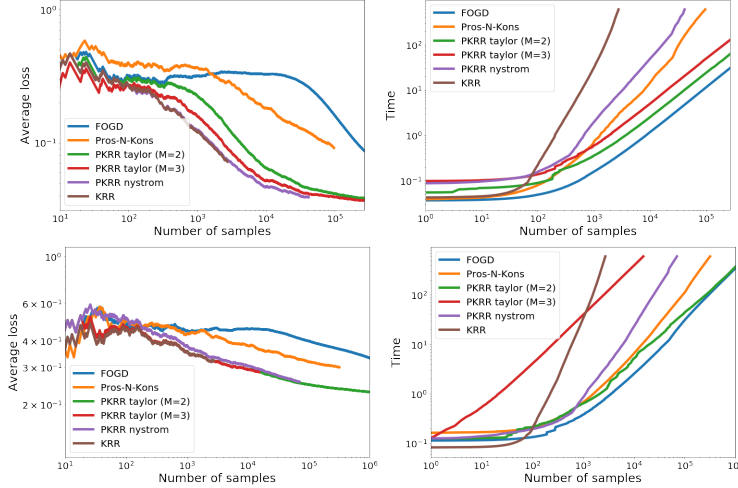

Figure 2: Average classification error and time on: (top) code-rna ($n = 2.7 \times 10^5$, $d = 8$); (bottom) SUSY ($n = 6 \times 10^6$, $d = 22$).

**Beforehand known features**   We may assume that the sequence of feature vectors $x_t$ is given in advance to the learner while only the outputs $y_t$ are sequentially revealed (see [GGHS18] or [BKM+15] for details). In this case, the complete dictionary $\mathcal{I}_n \subset \{x_1, \dots, x_n\}$ may be computed beforehand and PKAWV can be used with the fix subspace $\tilde{\mathcal{H}} = \mathrm{Span}(\phi(x), x \in \mathcal{I}_n)$. In this case, the regret upper-bound can be improved to $R_n \lesssim \lambda + d_{\mathrm{eff}}(\lambda) + \frac{n\mu}{\lambda}$ by removing a factor $m$ in the last term (see (9)).

**Corollary 8.** *Under the notation and assumptions of Corollary 7, PKAWV used with dictionary $\mathcal{I}_n$ and parameter $\mu = nm^{-1/\gamma}$ achieves with high probability*

$$
R_n \lesssim
\begin{cases}
n^{\frac{\gamma}{1+\gamma}} & \text{if } m \geq n^{\frac{2\gamma}{1+\gamma}} & \text{for } \lambda = n^{\frac{\gamma}{1+\gamma}} \\
nm^{-\frac{1}{2\gamma}} & \text{otherwise} & \text{for } \lambda = nm^{-\frac{1}{2\gamma}}
\end{cases} .
$$

*Furthermore, w.h.p. the dictionary is of size $|\mathcal{I}_n| \lesssim m$ leading to a per-round space and time complexity $O(m^2)$.*

The suboptimal rate due to a small dictionary is improved by a factor $\sqrt{m}$ compared to the "sequentially revealed features" setting. Furthermore, since $2\gamma/(1 + \gamma) < 1$ for all $\gamma \in (0, 1)$, the algorithm is able to recover the optimal rate $n^{\gamma/(1+\gamma)}$ for all $\gamma \in (0, 1)$ with a dictionary of sub-linear size $m \ll n$. We leave for future work the question whether there is really a gap between these two settings or if this gap from a suboptimality of our analysis.

## 4   Experiments

We empirically test PKAWV against several state-of-the-art algorithms for online kernel regression. In particular, we test our algorithms in (1) an adversarial setting [see Appendix G], (2) on large scale datasets. The following algorithms have been tested:

- Kernel-AWV for adversarial setting or Kernel Ridge Regression for i.i.d. real data settings;
- Pros-N-Kons [CLV17b];
- Fourier Online Gradient Descent (FOGD, [LHW+16]);
- PKAWV(or PKRR for real data settings) with Taylor expansions ($M \in \{2, 3, 4\}$)
- PKAWV(or PKRR for real data settings) with Nyström

The algorithms above have been implemented in python with numpy (the code for our algorithm is in Appendix H.2). The code necessary to reproduce the following experiments is available on GitHub at `https://github.com/Remjez/kernel-online-learning`. For most algorithms, we used hyperparameters from the respective papers. For all algorithms and all experiments, we set $\sigma = 1$ [except for SUSY where $\sigma = 4$, to match accuracy results from RCR17] and $\lambda = 1$. When

using KORS, we set $\mu = 1$, $\beta = 1$ and $\varepsilon = 0.5$ as in [CLV17b]. The number of random-features in FOGD is fixed to $1000$ and the learning rate $\eta$ is $1/\sqrt{n}$. All experiments have been done on a single desktop computer (Intel Core i7-6700) with a timeout of 5-min per algorithm. The results of the algorithms are only recorded up to this time.

**Large scale datasets.** The algorithms are evaluated on four datasets from UCI machine learning repository. In particular, `casp` (regression) and `ijcnn1`, `cod-rna`, `SUSY` (classification) [see Appendix G for `casp` and `ijcnn1`] ranging from $4 \times 10^4$ to $6 \times 10^6$ datapoints. For all datasets, we scaled $x$ in $[-1, 1]^d$ and $y$ in $[-1, 1]$. In Figs. 2 and 4 we show the average loss (square loss for regression and classification error for classification) and the computational costs of the considered algorithm.

In all the experiments PKAWV with $M = 2$ approximates reasonably well the performance of kernel forecaster and is usually very fast. We remark that using PKAWV $M = 2$ on the first million examples of `SUSY`, we achieve in 10 minutes on a single desktop, the same average classification error obtained with specific large scale methods for i.i.d. data [RCR17], although Kernel-AWV is using a number of features reduced by a factor 100 with respect to the one used in for FALKON in the same paper. Indeed they used $r = 10^4$ Nyström centers, while with $M = 2$ we used $r = 190$ features, validating empirically the effectiveness of the chosen features for the Gaussian kernel. This shows the effectiveness of the proposed approach for large scale machine learning problems with a moderate dimension $d$.

## Footnotes

[1] when $D = \infty$ we consider $\mathbb{R}^D$ as the space of squared summable sequences.

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
