[Supplementary Material]

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

# Supplementary material

The supplementary material is organized as follows:

- Appendix A starts with notations and useful identities that are used in the rest of the proofs
- Appendix B, C, D, E, and F contain the proofs mostly in order of appearance:
  - Appendix B: statement and proof of our main theorem on which are based most of our results.
  - Appendix C: proofs of Section 2 (Propositions 1 and 2)
  - Appendix D: proofs of section 3.1 (Theorem 3 and 4)
  - Appendix E: proofs of section 3.2 (Theorem 7 and Corollaries 7 and 8)
  - Appendix F: proofs of additional lemmas.
- Appendix G provides additional experimental results (adversarial simulated data and large-scale real datasets).
- Appendix H describes efficient implementations of our algorithms together with the Python code used for the experiments.

## A  Notations and relevant equations

In this section, we give notations and useful identities which will be used in following proofs. We recall that at time $t \geq 1$, the forecaster is given an input $x_t \in \mathcal{X} \subset \mathbb{R}^d$, chooses a prediction function $\widehat{f}_t \in \tilde{\mathcal{H}}_t \subset \mathcal{H}$, forecasts $\widehat{y}_t = \widehat{f}_t(x_t)$ and observes $\widehat{y}_t \in [-B, B]$. Moreover, $\mathcal{H}$ is the RKHS associated to the kernel $k : (x, x') \in \mathcal{X} \times \mathcal{X} = \phi(x)^\top \phi(x')$ for some feature map $\phi : \mathcal{X} \to \mathbb{R}^D$. We also define the following notations for all $t \geq 1$:

- $Y_t = (y_1, \ldots, y_t)^\top \in \mathbb{R}^t$ and $\widehat{Y}_t = (\widehat{y}_1, \ldots, \widehat{y}_t)^\top \in \mathbb{R}^t$
- $P_t : \mathcal{H} \to \tilde{\mathcal{H}}_t$ is the Euclidean projection on $\tilde{\mathcal{H}}_t$
- $C_t := \sum_{i=1}^t \phi(x_i) \otimes \phi(x_i)$ is the covariance operator at time $t \geq 1$;
- $A_t := C_t + \lambda I$ is the regularized covariance operator;
- $S_t : \mathcal{H} \to \mathbb{R}^t$ is the operator such that $[S_t f]_i = f(x_i) = \langle f, \phi(x_i) \rangle$ for any $f \in \mathcal{H}$;
- $L_t := f \in \mathcal{H} \mapsto \left\| Y_t - S_t f \right\|^2 + \lambda \|f\|^2$ is the regularized cumulative loss.

The prediction function of PKAWV at time $t \geq 1$ is defined (see Definition 4) as

$$\widehat{f}_t = \arg\min_{f \in \tilde{\mathcal{H}}_t} \left\{ \sum_{s=1}^{t-1} \left(y_s - f(x_s)\right)^2 + \lambda \|f\|^2 + f(x_t)^2 \right\} .$$

Standard calculation shows the equality

$$\widehat{f}_t = P_t \tilde{A}_t^{-1} P_t S_{t-1}^* Y_{t-1} . \tag{10}$$

We define also the best functions in the subspace $\tilde{\mathcal{H}}_t$ and $\tilde{\mathcal{H}}_{t+1}$ at time $t \geq 1$,

$$\widehat{g}_{t+1} = \arg\min_{f \in \tilde{\mathcal{H}}_t} \{L_t(f)\} = P_t \tilde{A}_t^{-1} P_t S_t^* Y_t , \tag{11}$$

$$\tilde{g}_{t+1} = \arg\min_{f \in \tilde{\mathcal{H}}_{t+1}} \{L_t(f)\} = P_{t+1}(P_{t+1} C_t P_{t+1} + \lambda I)^{-1} P_{t+1} S_t^* Y_t , \tag{12}$$

and the best function in the whole space $\mathcal{H}$

$$\widehat{h}_{t+1} = \arg\min_{f \in \mathcal{H}} \{L_t(f)\} = A_t^{-1} S_t^* Y_t . \tag{13}$$

# B   Main theorem (statement and proof)

In this appendix, we provide a general upper-bound on the regret of PKAWV that is valid for any sequence of projections $P_1, ..., P_n$ associated with the sequence $\tilde{\mathcal{H}}_1, \ldots, \tilde{\mathcal{H}}_n$. Many of our results will be corollaries of the following theorem for specific sequences of projections.

**Theorem 9.** *Let $\tilde{\mathcal{H}}_1, \ldots, \tilde{\mathcal{H}}_n$ be a sequence of linear subspaces of $\mathcal{H}$ associated with projections $P_1, \ldots, P_n \in \mathcal{H} \rightarrow \mathcal{H}$. PKAWV with regularization parameter $\lambda > 0$ satisfies the following upper-bound on the regret: for all $f \in \mathcal{H}$*

$$R_n(f) \leq \sum_{t=1}^{n} y_t^2 \left\langle \tilde{A}_t^{-1} P_t \phi(x_t), P_t \phi(x_t) \right\rangle + (\mu_t + \lambda) \frac{\mu_t t B^2}{\lambda} ,$$

*for any sequence $(x_1, y_1), \ldots, (x_n, y_n) \in \mathcal{X} \times [-B, B]$ and where $\mu_t := \left\| (P_{t+1} - P_t) C_t^{1/2} \right\|^2$ and $P_{n+1} := I$.*

*Proof.* Let $f \in \mathcal{H}$. By definition of $\widehat{h}_{n+1}$ (see (13)), we have $L_n(\widehat{h}_{n+1}) \leq L_n(f)$ which implies by definition of $L_n$ that

$$\left\| Y_n - S_n \widehat{h}_{n+1} \right\|^2 - \left\| Y_n - S_n f \right\|^2 \leq \lambda \|f\|^2 - \lambda \|\widehat{h}_{n+1}\|^2 . \tag{14}$$

Now, the regret can be upper-bounded as

$$R_n(f) \stackrel{(1)}{:=} \sum_{t=1}^{n} (y_t - \widehat{y}_t)^2 - \sum_{t=1}^{n} \left( y_t - f(x_t) \right)^2 \tag{15}$$

$$= \left\| Y_n - \widehat{Y}_n \right\|^2 - \left\| Y_n - S_n f \right\|^2$$

$$\stackrel{(14)}{\leq} \left\| Y_n - \widehat{Y}_n \right\|^2 - \left\| Y_n - S_n \widehat{h}_{n+1} \right\|^2 + \lambda \|f\|^2 - \lambda \|\widehat{h}_{n+1}\|^2$$

$$\leq \lambda \|f\|^2 + \underbrace{\left\| Y_n - \widehat{Y}_n \right\|^2 - \left\| Y_n - S_n \widehat{g}_{n+1} \right\|^2 - \lambda \|\widehat{g}_{n+1}\|^2}_{Z_1} \tag{16}$$

$$+ \underbrace{\left\| Y_n - S_n \widehat{g}_{n+1} \right\|^2 + \lambda \|\widehat{g}_{n+1}\|^2 - \left\| Y_n - S_n \widehat{h}_{n+1} \right\|^2 - \lambda \|\widehat{h}_{n+1}\|^2}_{\Omega(n+1)}$$

The first term $Z_1$ mainly corresponds to the estimation error of the algorithm: the regret incurred with respect to the best function in the approximation space $\tilde{\mathcal{H}}_n$. It also includes an approximation error due to the fact that the algorithm does not use $\tilde{\mathcal{H}}_n$ but the sequence of approximation $\tilde{\mathcal{H}}_1, \ldots, \tilde{\mathcal{H}}_n$. The second term $\Omega(n+1)$ corresponds to the approximation error of $\mathcal{H}$ by $\tilde{\mathcal{H}}_n$. Our analysis will focus on upper-bounding both of these terms separately.

**Part 1. Upper-bound of the estimation error $Z_1$.**   Using a telescoping argument together with the convention $L_0(\widehat{g}_1) = 0$, we have

$$\left\| Y_n - S_n \widehat{g}_{n+1} \right\|^2 + \lambda \|\widehat{g}_{n+1}\|^2 = L_n(\widehat{g}_{n+1}) = \sum_{t=1}^{n} L_t(\widehat{g}_{t+1}) - L_{t-1}(\widehat{g}_t) .$$

Substituted into the definition of $Z_1$ (see (16)), the latter can be rewritten as

$$Z_1 = \sum_{t=1}^{n} \left[ (y_t - \widehat{y}_t)^2 + L_{t-1}(\widehat{g}_t) - L_t(\widehat{g}_{t+1}) \right]$$

$$= \sum_{t=1}^{n} \Big[ \underbrace{(y_t - \widehat{y}_t)^2 + L_{t-1}(\tilde{g}_t) - L_t(\widehat{g}_{t+1})}_{Z(t)} + \underbrace{L_{t-1}(\widehat{g}_t) - L_{t-1}(\tilde{g}_t)}_{\Omega(t)} \Big] . \tag{17}$$

where $\tilde{g}_t = P_t (P_t C_{t-1} P_t + \lambda I)^{-1} P_t S_{t-1}^* Y_{t-1}$ is obtained by substituting $P_t$ with $P_{t-1}$ in the definition of $\widehat{g}_t$. Note that with the convention $P_{n+1} = I$ the second term $\Omega(t)$ matches the definition

of $\Omega(n+1)$ of (16) since $\tilde{g}_{n+1} = A_n^{-1} S_n^* Y_n = \widehat{h}_{n+1}$. In the rest of the first part we focus on upper-bounding the terms $Z(t)$. The approximation terms $\Omega(t)$ will be bounded in the next part.

Now, we remark that by expanding the square norm

$$L_t(f) = \|Y_t\|^2 - 2Y_t^\top S_t f + \left\|S_t f\right\|^2 + \lambda\|f\|^2 = \|Y_t\|^2 - 2Y_t^\top S_t f + \langle f, C_t f \rangle + \lambda\|f\|^2$$
$$= \|Y_t\|^2 - 2Y_t^\top S_t f + \langle f, A_t f \rangle , \quad (18)$$

where for the second equality, we used

$$\left\|S_t f\right\|^2 = \sum_{t=1}^n f(x_t)^2 = \sum_{t=1}^n \langle f, \phi(x_t) \rangle^2 = \sum_{t=1}^n \langle f, \phi(x_t) \otimes \phi(x_t) f \rangle = \langle f, C_t f \rangle .$$

Substituting $\widehat{g}_{t+1}$ into (18) we get

$$L_t(\widehat{g}_{t+1}) = \|Y_t\|^2 - 2Y_t^\top S_t \widehat{g}_{t+1} + \langle \widehat{g}_{t+1}, A_t \widehat{g}_{t+1} \rangle . \quad (19)$$

But, since $\widehat{g}_{t+1} \in \tilde{H}_t$, we have $\widehat{g}_{t+1} = P_t \widehat{g}_{t+1}$ which yields

$$Y_t^\top S_t \widehat{g}_{t+1} = Y_t^\top S_t P_t \widehat{g}_{t+1} = Y_t^\top S_t \tilde{A}_t^{-1} \tilde{A}_t P_t \widehat{g}_{t+1} .$$

Then, using that $\tilde{A}_t P_t = (P_t C_t P_t + \lambda I) P_t = P_t A_t P_t$, we get

$$Y_t^\top S_t \widehat{g}_{t+1} = \underbrace{Y_t^\top S_t P_t \tilde{A}_t^{-1} P_t}_{\widehat{g}_{t+1}^\top} A_t \widehat{g}_{t+1} = \langle \widehat{g}_{t+1}, A_t \widehat{g}_{t+1} \rangle .$$

Thus, combining with (19) we get

$$L_t(\widehat{g}_{t+1}) = \|Y_t\|^2 - \langle \widehat{g}_{t+1}, A_t \widehat{g}_{t+1} \rangle .$$

Similarly, substituting $\tilde{g}_t$ into (18) and using $\tilde{g}_t \in \tilde{\mathcal{H}}_t$, we can show

$$L_{t-1}(\tilde{g}_t) = \|Y_{t-1}\|^2 - \langle \tilde{g}_t, A_{t-1} \tilde{g}_t \rangle .$$

Combining the last two equations implies

$$L_{t-1}(\tilde{g}_t) - L_t(\widehat{g}_{t+1}) = -y_t^2 + \langle \widehat{g}_{t+1}, A_t \widehat{g}_{t+1} \rangle - \langle \tilde{g}_t, A_{t-1} \tilde{g}_t \rangle . \quad (20)$$

Furthermore, using the definition of $\widehat{g}_{t+1}$, we have

$$P_t A_t \widehat{g}_{t+1} = P_t A_t P_t \tilde{A}_t^{-1} P_t S_t^* Y_t = P_t \tilde{A}_t \tilde{A}_t^{-1} P_t S_t^* Y_t = P_t S_t^* Y_t .$$

The same calculation with $\tilde{g}_t$ yields

$$P_t A_{t-1} \tilde{g}_t = P_t (C_{t-1} + \lambda I) P_t (P_t C_{t-1} P_t + \lambda I)^{-1} S_{t-1}^* Y_{t-1}$$
$$= P_t (P_t C_{t-1} P_t + \lambda I)(P_t C_{t-1} P_t + \lambda I)^{-1} S_{t-1}^* Y_{t-1} = P_t S_{t-1}^* Y_{t-1} . \quad (21)$$

Together with the previous equality, it entails

$$P_t A_t \widehat{g}_{t+1} - P_t A_{t-1} \tilde{g}_t = P_t (S_t^* Y_t - S_{t-1}^* Y_{t-1}) = y_t P_t \phi(x_t) . \quad (22)$$

Then, because $\widehat{f}_t \in \tilde{\mathcal{H}}_t$, we have $\widehat{y}_t = \widehat{f}_t(x_t) = \left\langle \widehat{f}_t, \phi(x_t) \right\rangle = \left\langle \widehat{f}_t, P_t \phi(x_t) \right\rangle$. This yields

$$(y_t - \widehat{y}_t)^2 = y_t^2 - 2y_t \widehat{y}_t + \widehat{y}_t^2$$
$$= y_t^2 - 2\left\langle \widehat{f}_t, y_t P_t \phi(x_t) \right\rangle + \left\langle \widehat{f}_t, \phi(x_t) \otimes \phi(x_t) \widehat{f}_t \right\rangle$$
$$\overset{(22)}{\leq} y_t^2 - 2\left\langle \widehat{f}_t, P_t A_t \widehat{g}_{t+1} - P_t A_{t-1} \tilde{g}_t \right\rangle + \left\langle \widehat{f}_t, \phi(x_t) \otimes \phi(x_t) \widehat{f}_t \right\rangle$$
$$= y_t^2 - 2\left\langle \widehat{f}_t, A_t \widehat{g}_{t+1} - A_{t-1} \tilde{g}_t \right\rangle + \left\langle \widehat{f}_t, (A_t - A_{t-1}) \widehat{f}_t \right\rangle , \quad (23)$$

where the last equality uses $\widehat{f}_t \in \tilde{\mathcal{H}}_t$ and that $A_t - A_{t-1} = \phi(x_t) \otimes \phi(x_t)$.

Putting equations (20) and (23) together, we get

$$
\begin{aligned}
Z(t) \quad &\overset{(17)}{=} \quad (y_t - \widehat{y}_t)^2 + L_{t-1}(\tilde{g}_t) - L_t(\widehat{g}_{t+1}) \\[4pt]
&\overset{(20)+(23)}{\leq} \quad \left( \langle \widehat{g}_{t+1}, A_t \widehat{g}_{t+1} \rangle - 2 \left\langle \widehat{f}_t, A_t \widehat{g}_{t+1} \right\rangle + \left\langle \widehat{f}_t, A_t \widehat{f}_t \right\rangle \right) \\[4pt]
&\qquad - \left( \langle \tilde{g}_t, A_{t-1} \widehat{g}_t \rangle - 2 \left\langle P_{t-1} \widehat{f}_t, A_{t-1} \tilde{g}_t \right\rangle + \left\langle \widehat{f}_t, A_{t-1} \widehat{f}_t \right\rangle \right) \\[4pt]
&= \quad \left\langle \widehat{g}_{t+1} - \widehat{f}_t, A_t (\widehat{g}_{t+1} - \widehat{f}_t) \right\rangle - \underbrace{\left\langle \widehat{f}_t - \tilde{g}_t, A_{t-1}(\widehat{f}_t - \tilde{g}_t) \right\rangle}_{\geq 0} \\[4pt]
&\leq \quad \left\langle \widehat{g}_{t+1} - \widehat{f}_t, \tilde{A}_t(\widehat{g}_{t+1} - \widehat{f}_t) \right\rangle \\[4pt]
&\overset{(10)+(11)}{=} \quad \left\langle P_t \tilde{A}_t^{-1} P_t (S_t^* Y_t - S_{t-1}^* Y_{t-1}), \tilde{A}_t P_t \tilde{A}_t^{-1} P_t (S_t^* Y_t - S_{t-1}^* Y_{t-1}) \right\rangle \\[4pt]
&= \quad y_t^2 \left\langle P_t \tilde{A}_t^{-1} P_t \phi(x_t), \tilde{A}_t P_t \tilde{A}_t^{-1} P_t \phi(x_t) \right\rangle \\[4pt]
&= \quad y_t^2 \left\langle \tilde{A}_t^{-1} P_t \phi(x_t), P_t \phi(x_t) \right\rangle
\end{aligned}
$$

where the last equality is because $P_t \tilde{A}_t = \tilde{A}_t P_t$ from the definition of $\tilde{A}_t := \tilde{C}_t + \lambda I$ with $\tilde{C}_t := P_t C_t P_t$.

Therefore, plugging back into (17), we have

$$
Z_1 \leq \sum_{t=1}^{n} y_t^2 \left\langle \tilde{A}_t^{-1} P_t \phi(x_t), P_t \phi(x_t) \right\rangle + \Omega(t) \,, \tag{24}
$$

where we recall that $\Omega(t) := L_{t-1}(\widehat{g}_t) - L_{t-1}(\tilde{g}_t)$.

**Part 2. Upper-bound of the approximation terms $\Omega(t)$.** We recall that we use the convention $P_{n+1} = I$ which does not change the algorithm. Let $t \geq 1$, expending the square losses we get

$$
\begin{aligned}
\Omega(t+1) &= \sum_{s=1}^{t} \left[ (\widehat{g}_{t+1}(x_s) - y_s)^2 - (\tilde{g}_{t+1}(x_s) - y_s)^2 + \lambda \|\widehat{g}_{t+1}\|^2 - \lambda \|\tilde{g}_{t+1}\|^2 \right] \\[4pt]
&= \sum_{s=1}^{t} \big[ \cancel{y_s^2} - 2 \langle \widehat{g}_{t+1}, y_s \phi(x_s) \rangle + \langle \widehat{g}_{t+1}, \phi(x_s) \otimes \phi(x_s) \widehat{g}_{t+1} \rangle \\[4pt]
&\qquad - \cancel{y_s^2} + 2 \langle \tilde{g}_{t+1}, y_s \phi(x_s) \rangle - \langle \tilde{g}_{t+1}, \phi(x_s) \otimes \phi(x_s) \tilde{g}_{t+1} \rangle + \lambda \|\widehat{g}_{t+1}\|^2 - \lambda \|\tilde{g}_{t+1}\|^2 \big] \\[4pt]
&= 2 \langle \tilde{g}_{t+1} - \widehat{g}_{t+1}, S_t^* Y_t \rangle + \langle \widehat{g}_{t+1}, A_t \widehat{g}_{t+1} \rangle - \langle \tilde{g}_{t+1}, A_t \tilde{g}_{t+1} \rangle
\end{aligned}
$$

Since both $\tilde{g}_{t+1}$ and $\widehat{g}_{t+1}$ belong to $\tilde{\mathcal{H}}_{t+1}$, we have

$$
\Omega(t+1) = 2 \langle \tilde{g}_{t+1} - \widehat{g}_{t+1}, P_{t+1} S_t^* Y_t \rangle + \langle \widehat{g}_{t+1}, A_t \widehat{g}_{t+1} \rangle - \langle \tilde{g}_{t+1}, A_t \tilde{g}_{t+1} \rangle \,,
$$

which using that $P_{t+1} S_t^* Y_t = P_{t+1} A_t \tilde{g}_{t+1}$ by Equality (21) yields

$$
\begin{aligned}
\Omega(t+1) &= 2 \langle \tilde{g}_{t+1} - \widehat{g}_{t+1}, P_{t+1} A_t \tilde{g}_{t+1} \rangle + \langle \widehat{g}_{t+1}, A_t \widehat{g}_{t+1} \rangle - \langle \tilde{g}_{t+1}, A_t \tilde{g}_{t+1} \rangle \\[4pt]
&= 2 \langle \tilde{g}_{t+1} - \widehat{g}_{t+1}, A_t \tilde{g}_{t+1} \rangle + \langle \widehat{g}_{t+1}, A_t \widehat{g}_{t+1} \rangle - \langle \tilde{g}_{t+1}, A_t \tilde{g}_{t+1} \rangle \\[4pt]
&= -2 \langle \widehat{g}_{t+1}, A_t \tilde{g}_{t+1} \rangle + \langle \widehat{g}_{t+1}, A_t \widehat{g}_{t+1} \rangle + \langle \tilde{g}_{t+1}, A_t \tilde{g}_{t+1} \rangle \\[4pt]
&= \langle \tilde{g}_{t+1} - \widehat{g}_{t+1}, A_t (\tilde{g}_{t+1} - \widehat{g}_{t+1}) \rangle \,.
\end{aligned}
$$

Let us denote $B_t = P_{t+1} A_t P_{t+1}$. Then, remarking that $\widehat{g}_{t+1} = P_t \tilde{A}_t^{-1} P_t A_t \tilde{g}_{t+1}$ and that $(P_{t+1} - P_t \tilde{A}_t^{-1} P_t A_t) P_t = 0$, we have

$$
\begin{aligned}
\Omega(t+1) &= \left\langle (P_{t+1} - P_t \tilde{A}_t^{-1} P_t A_t) \tilde{g}_{t+1}, A_t (P_{t+1} - P_t \tilde{A}_t^{-1} P_t A_t) \tilde{g}_{t+1} \right\rangle \\[4pt]
&= \left\langle (P_{t+1} - P_t \tilde{A}_t^{-1} P_t B_t) \tilde{g}_{t+1}, B_t (P_{t+1} - P_t \tilde{A}_t^{-1} P_t B_t) \tilde{g}_{t+1} \right\rangle \\[4pt]
&= \left\| B_t^{1/2} (P_{t+1} - P_t \tilde{A}_t^{-1} P_t B_t) \tilde{g}_{t+1} \right\|^2 \\[4pt]
&= \left\| B_t^{1/2} (P_{t+1} - P_t \tilde{A}_t^{-1} P_t B_t)(P_{t+1} - P_t) \tilde{g}_{t+1} \right\|^2 \\[4pt]
&\leq \left\| B_t^{1/2} (P_{t+1} - P_t \tilde{A}_t^{-1} P_t B_t) \right\|^2 \left\| (P_{t+1} - P_t) \tilde{g}_{t+1} \right\|^2 \,. \tag{25}
\end{aligned}
$$

We now upper-bound the two terms of the right-hand-side. For the first one, we use that

$$\left\| P_{t+1} - B_t^{1/2} P_t \tilde{A}_t^{-1} P_t B_t^{1/2} \right\|^2$$

$$= \left\| P_{t+1} - 2 B_t^{1/2} P_t \tilde{A}_t^{-1} P_t B_t^{1/2} + B_t^{1/2} P_t \tilde{A}_t^{-1} \underbrace{P_t B_t^{1/2} B_t^{1/2} P_t \tilde{A}_t^{-1}}_{P_t} P_t B_t^{1/2} \right\|$$

$$= \left\| P_{t+1} - B_t^{1/2} P_t \tilde{A}_t^{-1} P_t B_t^{1/2} \right\|^2 \in \{0, 1\}, \tag{26}$$

where in the second equality we used that $P_t B_t^{1/2} B_t^{1/2} P_t \tilde{A}_t^{-1} = P_t B_t P_t \tilde{A}_t^{-1} = P_t \tilde{A}_t \tilde{A}_t^{-1} = P_t$.
Therefore, using that $B_t^{1/2} P_{t+1} = P_{t+1} B_t^{1/2}$ we get

$$\| B_t^{1/2} (P_{t+1} - P_t \tilde{A}_t^{-1} P_t B_t) \|^2 = \left\| B_t^{1/2} \left[ (P_{t+1} - P_t \tilde{A}_t^{-1} P_t B_t)(P_{t+1} - P_t) \right] \right\|^2$$

$$= \left\| (P_{t+1} - B_t^{1/2} s P_t \tilde{A}_t^{-1} P_t B_t^{1/2}) B_t^{1/2} (P_{t+1} - P_t) \right\|^2$$

$$\leq \left\| P_{t+1} - B_t^{1/2} P_t \tilde{A}_t^{-1} P_t B_t^{1/2} \right\|^2 \left\| B_t^{1/2} (P_{t+1} - P_t) \right\|^2$$

$$\overset{(26)}{\leq} \left\| B_t^{1/2} (P_{t+1} - P_t) \right\|^2$$

$$\leq \left\| C_t^{1/2} (P_{t+1} - P_t) \right\|^2 + \lambda$$

$$\leq \mu_t + \lambda,$$

where $\mu_t := \left\| (P_{t+1} - P_t) C_t^{1/2} \right\|^2$. Plugging back into (25), this yields

$$\Omega(t+1) \leq (\mu_t + \lambda) \| (P_{t+1} - P_t) \tilde{g}_{t+1} \|^2. \tag{27}$$

Then, substituting $\tilde{g}_{t+1}$ with its definition and using $\| Y_t \|^2 \leq t B^2$, we get

$$\| (P_{t+1} - P_t) \tilde{g}_{t+1} \|^2 \quad = \quad \| (P_{t+1} - P_t) A_t^{-1} S_t^* Y_t \|^2$$

$$\overset{\text{(Cauchy-Schwarz)}}{\leq} \quad \| (P_{t+1} - P_t) A_t^{-1} S_t^* \|^2 \| Y_t \|^2$$

$$\leq \quad t B^2 \| (P_{t+1} - P_t) A_t^{-1} S_t^* S_t A_t^{-1} (P_{t+1} - P_t) \|$$

$$\overset{(C_t = S_t^* S_t)}{=} \quad t B^2 \| (P_{t+1} - P_t) A_t^{-1} C_t A_t^{-1} (P_{t+1} - P_t) \|.$$

Because $C_t$ and $A_t = C_t + \lambda I$ are co-diagonalizable, we have

$$C_t^{1/2} A_t^{-1} = A_t^{-1} C_t^{1/2},$$

which, together with $\| A_t^{-2} \| \leq 1/\lambda^2$ leads to

$$\| (P_{t+1} - P_t) \tilde{g}_{t+1} \|^2 \quad \leq \quad t B^2 \| (P_{t+1} - P_t) C_t^{1/2} A_t^{-2} C_t^{1/2} (P_{t+1} - P_t) \|$$

$$\leq \quad \frac{t B^2}{\lambda^2} \| (P_{t+1} - P_t) C_t^{1/2} \|^2$$

$$= \quad \frac{t \mu_t B^2}{\lambda^2}.$$

Therefore, Inequality (27) concludes the proof of the second part

$$\Omega(t+1) \leq (\mu_t + \lambda) \frac{t \mu_t B^2}{\lambda^2}. \tag{28}$$

**Conclusion of the proof.** Combining (16), (24), and (28), we obtain

$$R_n(f) \quad \leq \quad \sum_{t=1}^{n} y_t^2 \left\langle \tilde{A}_t^{-1} P_t \phi(x_t), P_t \phi(x_t) \right\rangle + \sum_{t=1}^{n+1} \Omega(t)$$

$$\leq \quad \sum_{t=1}^{n} y_t^2 \left\langle \tilde{A}_t^{-1} P_t \phi(x_t), P_t \phi(x_t) \right\rangle + \sum_{t=1}^{n+1} (\mu_{t-1} + \lambda) \frac{(t-1) \mu_{t-1} B^2}{\lambda^2},$$

which concludes the proof of the theorem. $\qquad\square$

# C    Proofs of Section 2 (Kernel-AWV)

## C.1    Proof of Proposition 1

First, remark that Kernel-AWV corresponds to PKAWV with $\tilde{\mathcal{H}}_t = \mathcal{H}$ and thus $P_t = I$ for all $t \geq 1$. Therefore, applying Theorem 9 with $P_t = I$ yields the regret bound,

$$R_n(f) \leq \lambda \|f\|^2 + \sum_{t=1}^{n} \left\langle A_t^{-1} \phi(x_t), \phi(x_t) \right\rangle_{\mathcal{H}} ,$$

for all $f \in \mathcal{H}$. The rest of the proof consists in upper-bounding the second term in the right hand side. Remarking that $A_t = A_{t-1} + \phi(x_t) \otimes \phi(x_t)$ and applying Lemma 10 stated below we have

$$\left\langle A_t^{-1} \phi(x_t), \phi(x_t) \right\rangle_{\mathcal{H}} = 1 - \frac{\det(A_{t-1}/\lambda)}{\det(A_t/\lambda)} .$$

It is worth pointing out that $\det(A_t/\lambda)$ is well defined since $A_t = I + C_t$ with $C_t = \sum_{s=1}^{t} \phi(x_s) \otimes \phi(x_s)$ at most of rank $t \geq 0$. Then we use $1 - u \leq \log(1/u)$ for $u > 0$ which yields

$$\left\langle A_t^{-1} \phi(x_t), \phi(x_t) \right\rangle_{\mathcal{H}} \leq \log \frac{\det(A_t/\lambda)}{\det(A_{t-1}/\lambda)} .$$

Summing over $t = 1, \ldots, n$, using $A_0 = \lambda I$ and $A_n = \lambda I + C_n$ we get

$$\sum_{t=1}^{n} \left\langle A_t^{-1} \phi(x_t), \phi(x_t) \right\rangle_{\mathcal{H}} \leq \log \left( \det \left( I + \frac{C_n}{\lambda} \right) \right)$$

$$= \sum_{k=1}^{\infty} \log \left( 1 + \frac{\lambda_k(C_n)}{\lambda} \right) ,$$

which concludes the proof.

The following Lemma is a standard result of online matrix theory (see Lemma 11.11 of [CBL06]).

**Lemma 10.** *Let $V : \mathcal{H} \to \mathcal{H}$ be a linear operator. Let $u \in \mathcal{H}$ and let $U = V - u \otimes u$. Then,*

$$\left\langle V^{-1} u, u \right\rangle_{\mathcal{H}} = 1 - \frac{\det(U)}{\det(V)} .$$

## C.2    Proof of Proposition 2

Using that for $x > 0$
$$\log(1 + x) \leq \frac{x}{x+1} (1 + \log(1 + x)) ,$$
and denoting by $a(\lambda)$ the quantity $a(s, \lambda) := 1 + \log(1 + s/\lambda)$, we get for any $n \geq 1$

$$\log \left( 1 + \frac{\lambda_k(K_{nn})}{\lambda} \right) \leq \frac{\lambda_k(K_{nn})}{\lambda + \lambda_k(K_{nn})} a(\lambda_k(K_{nn}), \lambda).$$

Therefore, summing over $k \geq 1$ and denoting by $\lambda_1$ the largest eigenvalue of $K_{nn}$

$$\sum_{k=1}^{n} \log \left( 1 + \frac{\lambda_k(K_{nn})}{\lambda} \right) \leq a(\lambda_1, \lambda) \sum_{k=1}^{n} \frac{\lambda_k(K_{nn})}{\lambda + \lambda_k(K_{nn})} \tag{29}$$

$$= a(\lambda_1, \lambda) \operatorname{Tr} \left( K_{nn}(K_{nn} + \lambda I)^{-1} \right)$$

$$= a(\lambda_1, \lambda) d_{\text{eff}}(\lambda)$$

where the last equality is from the definition of $d_{\text{eff}}(\lambda)$. Combining with Proposition 1, substituting $a$ and upper-bounding

$$\lambda_1(K_{nn}) \leq \operatorname{Tr}(K_{nn}) = \sum_{t=1}^{n} \|\phi(x_t)\|^2 \leq n\kappa^2$$

concludes the proof.

# D  Proofs of Section 3.1 (PKAWV with Taylor's expansion)

## D.1  Proof of Theorem 3

Applying Theorem 9 with a fix projection $P$ and following the lines of the proof of Proposition 1 we get

$$R_n(f) \le \lambda \|f\|^2 + B^2 \sum_{j=1}^n \log\left(1 + \frac{\lambda_j(PC_nP)}{\lambda}\right) + (\mu + \lambda)\frac{n\mu B^2}{\lambda^2},$$

where $\mu = \|(I-P)C_n^{1/2}\|^2$. Moreover we have for all $i = 1, \dots, n$ using that $\tilde{C}_n = PC_nP = PS_nS_n^*P$, we have

$$\lambda_i(\widetilde{C}_n) = \lambda_i(PC_nP) = \lambda_i(PS_n^*S_nP) = \lambda_i(S_nPPS_n^*) = \lambda_i(S_nPS_n^*) \le \lambda_i(K_{nn}).$$

## D.2  Proof of Theorem 4

To apply our Thm. 3, we need first (1) to recall that the functions $g_k$, $k \in \mathbb{N}_0^d$ are in $\mathcal{H}$, (2) to show that they can approximate perfectly the kernel and (3) to quantify the approximation error of $G_M$ for the kernel function. First we recall some important existing results about the considered set of functions. For completeness, we provide self-contained (and often shorter and simplified) proofs of the following lemmas in Appendix F.

The next lemma states that $g_k$ with $k \in \mathbb{N}$ is an orthonormal basis for $\mathcal{H}$ induced by the Gaussian kernel.

**Lemma 11** ([SHS06]). *For any $k, k' \in \mathbb{N}_0^d$,*

$$g_k \in \mathcal{H}, \quad \|g_k\|_{\mathcal{H}} = 1, \quad \langle g_k, g_{k'}\rangle_{\mathcal{H}} = \mathbb{1}_{k=k'}.$$

Note that byproduct of the lemma, we have that $G_M \subset \mathcal{H}$ and moreover that the matrix $Q$ is the identity, indeed $Q_{ij} = \langle g_{k_i}, g_{k_j}\rangle_{\mathcal{H}} = \mathbb{1}_{k_i=k_j}$. This means that the functions in $G_M$ are linearly independent. Moreover the fact that $Q = I_r$ further simplifies the computation of the embedding $\tilde{\phi}$ (see (38)) in the implementation of the algorithm.

The next lemma recalls the expansion of $k(x, x')$ in terms of the given basis.

**Lemma 12** ([CKS11]). *For any $x \in \mathcal{X}$,*

$$k(x, x') = \langle \phi(x), \phi(x')\rangle_{\mathcal{H}} = \sum_{k \in \mathbb{N}_0^d} g_k(x)g_k(x'). \tag{30}$$

Finally, next lemma provides approximation error of $k(x, x')$ in terms of the set of functions in $G_M$, when the data is contained in a ball or radius $R$.

**Lemma 13** ([CKS11]). *Let $R > 0$. For any $x, x' \in \mathbb{R}^d$ such that $\|x\|, \|x'\| \le R$ we have*

$$\left|k(x, x') - \sum_{g \in G_M} g(x)g(x')\right| \le \frac{(R/\sigma)^{2M+2}}{(M+1)!}. \tag{31}$$

Now we are ready to prove Thm. 4.

**Proof of point 1.** First, note that $r := |G_M|$, the cardinality of $G_M$, corresponds to the number of monomials of the polynomial $(1 + x_1 + \cdots + x_d)^d$, i.e. $r := |G_M| = \binom{M+d}{M}$. By recalling that $\binom{n}{k} \le (en/k)^k$ for any $n, k \in \mathbb{N}$, we have

$$r = \binom{M+d}{M} = \binom{M+d}{d} \le e^d(1 + M/d)^d.$$

We conclude the proof of the first point of the theorem, by considering that PKAWV used with the set of functions $G_M$ consists in running the online linear regression algorithm of [Vov01, AW01] with $r := |G_M|$ features (see Appendix H for details). It incurs thus a computational cost of $O(nr^2 + nrd)$ in time (no $r^3$ since we don't need to invert $Q$ which we have proven to be the identity matrix as consequence of Lemma 11) and $O(r^2)$ in memory.

**Proof of point 2.** By Lemma 11 we have that $G_M \subset \mathcal{H}$ and $Q = I_r$, so the functions in $G_M$ are linearly independent. Then we can apply Thm. 3 obtaining the regret bound in Eq. (5):

$$R_n(f) \leq \lambda \|f\|^2 + B^2 \sum_{j=1}^n \log\left(1 + \frac{\lambda_j(K_{nn})}{\lambda}\right) + B^2 \frac{(\mu + \lambda)n}{\lambda^2}\mu\,, \tag{32}$$

where $\mu := \left\|(I - P)C_n^{1/2}\right\|^2$ and $C_n := \sum_{t=1}^n \phi(x_t) \otimes \phi(x_t)$. The proof consists in upper-bounding the last approximation term $B^2 \frac{(\mu+\lambda)n}{\lambda^2}\mu$. We start by upper-bounding $\mu$ as follows

$$
\begin{aligned}
\mu := \left\|(I - P)C_n^{1/2}\right\|^2 &= \|(I - P)C_n(I - P)\| \\
&= \left\|(I - P)\sum_{t=1}^n \phi(x_t) \otimes \phi(x_t)(I - P)\right\| \\
&\leq \sum_{t=1}^n \|(I - P)\phi(x_t) \otimes \phi(x_t)(I - P)\| \\
&= \sum_{t=1}^n \|(I - P)\phi(x_t)\|^2 \\
&= \sum_{t=1}^n \langle(I - P)\phi(x_t), \phi(x_t)\rangle \\
&= \sum_{t=1}^n \langle\phi(x_t), \phi(x_t)\rangle - \langle P\phi(x_t), P\phi(x_t)\rangle \\
&= \sum_{t=1}^n k(x_t, x_t) - \|P\phi(x_t)\|^2\,,
\end{aligned}
$$

where we used that $\langle P\phi(x_t), \phi(x_t)\rangle = \langle P\phi(x_t), P\phi(x_t)\rangle$. Now, since by Lemma 11, the $g_k$ form an orthonormal basis of $\mathcal{H}$, we have that

$$\|P\phi(x_t)\|^2 = \sum_{g \in G_M} g(x_t)^2\,.$$

where we recall that $P$ the projection onto $G_M$. Therefore, by Lemma 13,

$$\mu \leq \frac{(R/\sigma)^{2M+2}n}{(M+1)!} \overset{\text{Stirling}}{\leq} \frac{ne^{-(M+1)\log\left(\frac{(M+1)\sigma^2}{eR^2}\right)}}{\sqrt{2\pi(M+1)}} \leq \frac{ne^{-(M+1)}}{\sqrt{2\pi(M+1)}} \overset{M \geq 1}{\leq} \frac{n}{9}e^{-M}\,, \tag{33}$$

where we used the fact that $n!$ is lower bounded by the Stirling approximation as $n! \geq \sqrt{2\pi n}e^{n\log\frac{n}{e}}$, for $n \in \mathbb{N}_0$ and that $M + 1 \geq e^2 R^2/\sigma^2$, so $\log\frac{M+1}{eR^2/\sigma^2} \geq 1$. Now, since $M \geq 2\log(n/(\lambda \wedge 1))$, we have $M \geq \log(n/\lambda)$ and thus

$$\mu \leq \frac{n}{9}e^{-M} \leq \frac{\lambda}{9} \leq \lambda.$$

Therefore, the approximation term in (32) is upper-bounded as

$$B^2 \frac{(\mu + \lambda)}{\lambda^2}\mu n \leq \frac{2B^2\mu n}{\lambda} \overset{(33)}{\leq} \frac{2B^2n^2e^{-M}}{9\lambda}$$

which using again $M \geq 2\log(n/(\lambda \wedge 1))$ entails

$$B^2 \frac{(\mu + \lambda)}{\lambda^2}\mu n \leq \frac{2}{9}B^2(\lambda \wedge \lambda^{-1}) \leq \frac{4B^2}{9}\log\left(1 + \frac{1}{\lambda}\right)\,, \tag{34}$$

where in the last inequality we used that $(\lambda \wedge \lambda^{-1})/2 \leq \log(1 + 1/\lambda)$ for any $\lambda > 0$. Now, since $\log(1 + x)$ is concave on $[0, \infty)$, by subadditivity

$$\sum_{j=1}^n \log\left(1 + \frac{\lambda_j(K_{nn})}{\lambda}\right) \geq \log\left(1 + \sum_{j=1}^n \frac{\lambda_j(K_{nn})}{\lambda}\right).$$

By definition of trace in terms of eigenvalues and of the diagonal of $K_{nn}$, we have

$$\sum_{j=1}^{n} \lambda_j(K_{nn}) = \text{Tr}(K_{nn}) = \sum_{j=1}^{n} k(x_j, x_j) = n,$$

where the last step is due to the fact that for the Gaussian kernel we have $k(x,x) = 1$, for any $x \in \mathcal{X}$. Then

$$B^2 \log\left(1 + \frac{1}{\lambda}\right) \leq B^2 \log\left(1 + \frac{n}{\lambda}\right) \leq B^2 \sum_{j=1}^{n} \log\left(1 + \frac{\lambda_j(K_{nn})}{\lambda}\right). \tag{35}$$

Plugging back into Inequality (34) and substituting into (32) concludes the proof of the Theorem.

# E  Proofs of Section 3.2 (PKAWV with Nyström projections)

## E.1  Proof of Theorem 6

The proof consists of a straightforward combination of Proposition 5 and Theorem 9. According to Proposition 5, with probability at least $1 - \delta$, we have for all $t \geq 1$,

$$\mu_t = \|(P_{t+1} - P_t)C_t^{1/2}\|^2 \leq \|(I - P_t)C_t^{1/2}\|^2 \mathbb{1}_{P_{t+1} \neq P_t} \leq \mu \mathbb{1}_{P_{t+1} \neq P_t},$$

with $|\mathcal{I}_n| \leq 9d_{\text{eff}}(\mu) \log(2n/\delta)^2$. Therefore, from Theorem 9, if $\mu \leq \lambda$, the regret is upper-bounded as

$$R_n(f) \leq \lambda\|f\|^2 + B^2 d_{\text{eff}}(\lambda) \log\left(e + \frac{en\kappa^2}{\lambda}\right) + 2\frac{\mu n(|\mathcal{I}_n| + 1)B^2}{\lambda}.$$

Furthermore, similarly to any online linear regression algorithm in a $m$-dimensional space, the efficient implementation of the algorithm (see Appendix H) requires $O(m^2)$ space and time per iteration, where $m = |\mathcal{I}_n|$ is the size of the dictionary. This concludes the proof of the theorem.

## E.2  Proof of Corollary 7

We recall that the notation $\lesssim$ denotes an approximate inequality which is up to logarithmic multiplicative terms and may depend on unexplained constants. Here, we only consider non-constant quantities $n$, $\lambda$, $m$ and $\mu$ and focus on the polynomial dependence on $n$. Keeping this in mind, the high-probability regret upper-bound provided by Theorem 6 can be rewritten as

$$R_n(f) \lesssim \lambda + \left(\frac{n}{\lambda}\right)^{\gamma} + \frac{\mu n|\mathcal{I}_n|}{\lambda}, \tag{36}$$

for all $f \in \mathcal{H}$. It only remains to optimize the parameters $\mu$ and $\lambda$. Choosing $\mu = d_{\text{eff}}^{-1}(m)$ ensures that the size of the dictionary is upper-bounded as $|\mathcal{I}_n| \lesssim d_{\text{eff}}(\mu) = m$.

Moreover, by assumption $m = d_{\text{eff}}(\mu) \leq \left(\frac{n}{\mu}\right)^{\gamma}$ and thus $\mu \leq nm^{-\frac{1}{\gamma}}$. Therefore, the regret is upper-bounded with high-probability as

$$R_n(f) \lesssim \lambda + \left(\frac{n}{\lambda}\right)^{\gamma} + \frac{n^2(m^{\frac{\gamma-1}{\gamma}} + 1)}{\lambda}. \tag{37}$$

Now, according to the value of $m$, two regimes are possible:

- If the dictionary is large enough, i.e., $m \geq n^{\frac{2\gamma}{1-\gamma^2}}$ then, once $\lambda$ is optimized, the last term of the right-hand side is negligible. The regret upper-bound consists then in optimizing $\lambda + (n/\lambda)^{\gamma}$ in $\lambda$ yielding to the choice $\lambda = n^{\frac{\gamma}{1+\gamma}}$. We get the upper-bound

$$R_n(f) \lesssim n^{\frac{\gamma}{\gamma+1}} + n^{\gamma} n^{-\frac{\gamma^2}{1+\gamma}} + n^2 n^{-\frac{\gamma}{\gamma+1}} n^{\frac{-2}{\gamma+1}} \lesssim n^{\frac{\gamma}{\gamma+1}},$$

  which recovers the optimal rate in this case.

- Otherwise, if $m \leq n^{\frac{2\gamma}{1-\gamma^2}}$, then the last term of the r.h.s. of (37) is predominant. The dictionary is too small to recover the optimal regret bound. The parameter $\lambda$ is optimizes the trade-off $\lambda + n^2 m^{(\gamma-1)/\gamma}/\lambda$ which leads to the choice $\lambda = nm^{\frac{1}{2}-\frac{1}{2\gamma}}$. The upper-bound on the regret is then

$$R_n(f) \lesssim nm^{\frac{\gamma-1}{2\gamma}} + m^{\frac{1-\gamma}{2}} + nm^{\frac{1-\gamma}{2\gamma}+\frac{\gamma-1}{\gamma}} \lesssim nm^{\frac{\gamma-1}{2\gamma}}.$$

This concludes the proof.

### E.3  Proof of Corollary 8

The proof follows the lines of the one of Theorem 6 and Corollary 7. However, here since the projections are fixed we can apply Theorem 3 instead of Theorem 9. This yields the high-probability regret upper-bound

$$R_n(f) \lesssim \lambda + \left(\frac{n}{\lambda}\right)^\gamma + \frac{\mu n}{\lambda},$$

which improves by a factor $|\mathcal{I}_n|$ the last term of the bound (36). The choice $\mu = d_{\mathrm{eff}}^{-1}(m)$ yields with high probability $|\mathcal{I}_n| \lesssim d_{\mathrm{eff}}(\mu) = m$ and $\mu \leq nm^{-\frac{1}{\gamma}}$ which entails

$$R_n(f) \lesssim \lambda + \left(\frac{n}{\lambda}\right)^\gamma + \frac{n^2 m^{-\frac{1}{\gamma}}}{\lambda}.$$

Similarly to Corollary 7 two regimes are possible. The size of the dictionary decides which term is preponderant in the above upper-bound:

- If $m \geq n^{\frac{2\gamma}{1+\gamma}}$ the dictionary is large enough to recover the optimal rate for the choice $\lambda = n^{\frac{\gamma}{1+\gamma}}$. Indeed it yields

$$R_n(f) \lesssim n^{\frac{\gamma}{\gamma+1}} + n^\gamma n^{-\frac{\gamma^2}{1+\gamma}} + n^{\frac{2\gamma}{\gamma+1}} n^{-\frac{\gamma}{\gamma+1}} \lesssim n^{\frac{\gamma}{\gamma+1}}$$

- Otherwise $m \leq n^{\frac{2\gamma}{1+\gamma}}$ and the choice $\lambda = n^{\frac{\gamma}{1+\gamma}}$ leads to

$$R_n(f) \lesssim n^{\frac{\gamma}{\gamma+1}} + n^\gamma n^{-\frac{\gamma^2}{1+\gamma}} + n^{\frac{2\gamma}{\gamma+1}} n^{-\frac{\gamma}{\gamma+1}} \lesssim n^{\frac{\gamma}{\gamma+1}}.$$

  The last inequality is due to $m^{\frac{1}{2}} \leq nm^{-\frac{1}{\gamma}+\frac{1}{2\gamma}}$ because $\gamma \leq 1$ and $m \leq n$.

## F  Proofs of additional lemmas

### F.1  Proof of Lemma 11

Recall the following characterization of scalar product for translation invariant kernels (i.e. $k(x, x') = v(x - x')$ for a $v : \mathbb{R}^d \to \mathbb{R}$) [see BTA11]

$$\langle f, g \rangle_{\mathcal{H}} = \int \frac{\mathcal{F}[f](\omega)\mathcal{F}[g](\omega)}{\mathcal{F}[v](\omega)},$$

where $\mathcal{F}[f]$ is the unitary Fourier transform of $f$. Let start from the one dimensional case and denote by $\mathcal{H}_0$ the Gaussian RKHS on $\mathbb{R}$. First note that when $d = 1$, we have $g_k = \psi_k$. Now, the Fourier transform of $\psi_k$ is $\mathcal{F}[\psi_k](\omega) = \frac{1}{\sqrt{k!}} H_k(x/\sigma^2)e^{-\omega^2/(2\sigma^2)}$, for any $k \in \mathbb{N}_0^d$, where $H_k(x)$ is the $k$-th Hermite polynomial [see OLBC10, Eq. 18.17.35 pag. 457], and $\mathcal{F}[v] = e^{-\omega^2/2}$, then, by the fact that Hermite are orthogonal polynomial with respect to $e^{-\omega^2/2}$ forming a complete basis, we have

$$\langle \psi_k, \psi_{k'} \rangle_{\mathcal{H}_0} = \frac{1}{k!} \int H_k(\omega) H_{k'}(\omega) e^{-\omega^2/2} d\omega = \mathbb{1}_{k=k'}.$$

The multidimensional case is straightforward since Gaussian is a product kernel, i.e. $k(x, x') = \prod_{i=1}^d k(x^{(i)}, x^{(i)})$ and $\mathcal{H} = \otimes_{i=1}^d \mathcal{H}_0$, so $\langle \otimes_{i=1}^d f_i, \otimes_{i=1}^d g_i \rangle_{\mathcal{H}} = \prod_{i=1}^d \langle f_i, g_i \rangle_{\mathcal{H}_0}$ [see Aro50]. Now, since $g_k = \otimes_{i=1}^d \psi_{k_i}$, we have $\langle g_k, g_{k'} \rangle_{\mathcal{H}} = \prod_{i=1}^d \langle \psi_{k_i}, \psi_{k'_i} \rangle_{\mathcal{H}_0} = \mathbb{1}_{k=k'}.$

### F.2 Proof of Lemma 12

First, for $j \in \mathbb{N}_0$ define

$$Q_j(x, x') := e^{-\frac{\|x\|^2}{2\sigma^2} - \frac{\|x'\|^2}{2\sigma^2}} \frac{(x^\top x'/\sigma^2)^j}{j!}.$$

First note that, by multinomial expansion of $(x^\top x')^j$,

$$Q_j(x, x') = \frac{e^{-\frac{\|x\|^2 + \|x'\|^2}{2\sigma^2}}}{\sigma^{2j} j!} \sum_{|t|=j} \binom{j}{t_1 \dots t_d} \prod_{i=1}^d (x^{(i)})^{t_i} (x'^{(i)})^{t_i}$$

$$= \sum_{|t|=j} g_t(x) g_t(x').$$

Now note that, by Taylor expansion of $e^{x^\top x'/\sigma^2}$ we have

$$k(x, x') = \sum_{j=0}^\infty Q_j(x, x') = \sum_{j=0}^\infty \sum_{|t|=j} g_t(x) g_t(x')$$

$$= \sum_{k \in \mathbb{N}_0^d} g_k(x) g_k(x').$$

Finally, with $\phi$ defined as above, and the fact that $g_k$ forms an orthonormal basis for $\mathcal{H}$, leads to

$$\langle \phi(x), \phi'(x) \rangle = \sum_{k \in \mathbb{N}_0^d} g_k(x) g_k(x') = k(x, x').$$

### F.3 Proof of Lemma 13

Here we use the same notation of the proof of Lemma 12. Since by Taylor expansion, we have that $k(x, x') = \sum_{j=0}^\infty Q_j(x, x')$, by mean value theorem for the function $f(s) = e^{s/\sigma^2}$, we have that there exists $c \in [0, x^\top x']$ such that

$$|k(x, x') - \sum_{j=0}^M Q_j(x, x')| = e^{-\frac{\|x\|^2 + \|x'\|^2}{2\sigma^2}} \frac{c^{M+1}}{(M+1)!} \frac{d^{M+1} e^{\frac{s}{\sigma^2}}}{ds^{M+1}}\big|_{s=c}$$

$$\leq \frac{(|x^\top x'|/\sigma^2)^{M+1}}{(M+1)!}$$

$$\leq \frac{(R/\sigma)^{2M+2}}{(M+1)!}$$

where the last step is obtained assuming $\|x\|, \|x'\| \leq R$. Finally note that, by definition of $G_M$,

$$\sum_{g \in G_M} g(x) g(x') = \sum_{|k| \leq M}^M g_k(x) g_k(x') = \sum_{j=0}^M Q_j(x, x').$$

## G  Additional experiments

**Additional large scale datasets** (cf. Figure 4). We provides results on two additional datasets from UCI machine learning repository : `casp` (regression) and `ijcnn1`. See section 4 for more details.

**Adversarial simulated data** (cf. Figure 3) In this experiment we produced the sequence $(x_t, y_t)_{t \in \mathbb{N}}$ adversarially on the regret function. In particular, given the learning algorithm, we use `scipy` as a greedy adversary i.e. at each step an optimization is done on the regret function to find $(x_t, y_t)$. On the right of Figure 3, we plot the simulations until $n = 80$, with $(x_t, y_t) \in [-1, 1]^d \times [-1, 1]$ where $d = 5$. We see that Kernel-AWV, which does not use any approximation, leads to the best regret.

Figure 3: Regret in adversarial setting.

Figure 4: Average loss and time on (top): regression casp ($n = 4.5 \times 10^4, \ d = 9$); (bottom) classification `ijcnn1` ($n = 1.5 \times 10^5, \ d = 22$).

Furthermore, PKAWV approximations converges very fast to the regret of Kernel-AWV when $M$ increases. The poor performance of Pros-N-Kons is likely because of its frequent restarts which is harmful when $n$ is small. On the contrary, FOGD has surprisingly good performance. We run the simulations up to $n = 80$ for the high computational cost required by the adversary (especially for algorithms like Kernel-AWV or Pros-N-Kons).

# H Efficient implementation of PKAWV

## H.1 Pseudo-code

Here, we detail how the formula (4) can be efficiently computed for the projections considered in Section 3.

**Fixed embedding**    We consider fix sub-spaces $\tilde{\mathcal{H}}_t = \tilde{\mathcal{H}}$ induced fixed by the span of a fixed set of functions $G = \{g_1, \ldots, g_r\} \subset \mathcal{H}$ as analyzed in Section 3.1. Let denote by $\tilde{\phi} : \mathcal{X} \to \mathbb{R}^r$ the map

$$\tilde{\phi}(x) = Q^{-1/2}v(x), \tag{38}$$

with $v(x) = (g_1(x), \ldots, g_r(x))$, and $Q \in \mathbb{R}^{r \times r}$ defined as $Q_{ij} = \langle g_i, g_j \rangle_{\mathcal{H}}$. Then, computing the prediction $\widehat{y}_t = \widehat{f}_t(x_t)$ of PKAWV with

$$\widehat{f}_t \in \underset{f \in \tilde{\mathcal{H}} = \mathrm{Span}(G)}{\mathrm{argmin}} \left\{ \sum_{s=1}^{t-1} \left(y_s - f(x_s)\right)^2 + \lambda \|f\|^2 + f(x_t)^2 \right\}$$

is equivalent to embedding $x_t$ in $\mathbb{R}^r$ via $\tilde{\phi}$ and then performing linear AWV of [AW01, Vov01] with $\widehat{y}_t = \widehat{w}_t^\top \tilde{\phi}(x_t)$

$$\widehat{w}_t \in \underset{w \in \mathbb{R}^r}{\mathrm{argmin}} \left\{ \sum_{s=1}^{t-1} \left(y_s - w^\top \tilde{\phi}(x_s)\right)^2 + \lambda \|w\|^2 + \left(w^\top \tilde{\phi}(x_t)\right)^2 \right\}.$$

This reduces the total computational complexity to $O(nr^2 + nrd + r^3)$ in time and $O(r^2)$ in space (see Algorithm 1 for an efficient implementation).

---

**Algorithm 1** PKAWV with fixed embedding

---

**Input**: $\lambda > 0$, $\tilde{\phi} : \mathcal{X} \to \mathbb{R}^r$ for $r \geq 1$
**Initialization**: $A_0^{-1} = \lambda^{-1}I_r$, $b_0 = 0$
**For** $t = 1, \ldots, n$

   – receive $x_t \in \mathcal{X}$

   – compute $v_t = \tilde{\phi}(x_t) \in \mathbb{R}^r$

   – update $A_t^{-1} = A_{t-1}^{-1} - \frac{(A_t^{-1}v_t)(A_t^{-1}v_t)^\top}{1 + v_t^\top A_t^{-1} v_t}$

   – predict $\widehat{y}_t = \tilde{\phi}(x_t)^\top A_t^{-1} b_{t-1}$

   – receive $y_t \in \mathbb{R}$

   – update $b_t = b_{t-1} + v_t y_t$

---

**Nyström projections**    Here, we detail how our algorithm can be efficiently implemented with Nyström projections as considered in section 3.2. If we implement naïvely this algorithm, we would compute $\alpha_t = (K_{t,m_t}^T K_{t,m_t} + \lambda K_{m_t,m_t})^{-1} K_{t,m_t}^T Y_t$ at each iteration. However, it would require $nd_{\mathrm{eff}}(\mu) + d_{\mathrm{eff}}(\mu)^3$ operations per iterations. We could have update this inverse with Sherman–Morrison formula and Woodbury formula. However, in practice it leads to numeric instability because the matrix can have small eigenvalues. Here we use a method described in [RCR15]. The idea is to use the cholesky decomposition and cholup which update the cholesky decomposition when adding a rank one matrix i.e. if $A_t = L_t^T L_t$ and $A_{t+1} = A_t + u_{t+1}u_{t+1}^T$ then $L_{t+1} = \mathrm{cholup}(L_t, u_{t+1}, '+')$. Updating the cholesky decomposition with cholup require only $d_{\mathrm{eff}}(\mu)^2$ operations. So, PKAWV with nyström has a $O(nd_{\mathrm{eff}}(\mu) + d_{\mathrm{eff}}(\mu)^2)$ time complexity per iterations.

**Algorithm 2** PKAWV with Nyström projections

**Input**: $\lambda, \mu, \beta > 0$,

**Initialization**: $d_1 =$

**For** $t = 1, \ldots, n$

    – receive $x_t \in \mathcal{X}$

    – compute $z_t$ with KORS

    – $K_t = (k(x_i, \tilde{x}_j))_{i \leq t, j \in \mathcal{I}_{t-1}}$

    – $\mathcal{I}_t = \mathcal{I}_{t-1}$

    – $a_t = (k(x_t, x_1), \ldots, k(x_t, x_t))$

    – $R_t = \text{cholup}(R_t, a_t, '+')$

    **If** $z_t = 1$

        – $\mathcal{I}_t = \mathcal{I}_t \cup \{t\}$

        – $K_t = (k(x_i, x_j))_{i \leq t, j \in \mathcal{I}_t}$

        – $b_t = (k(x_t, x_j))_{j \in \mathcal{I}_t}$

        – $c_t = K_{t-1}^T a_t + \lambda b_t$

        – $d_t = a_t^T a_t + \lambda k(x_t, x_t)$

        – $g_t = \sqrt{1 + d_t}$

        – $u_t = (c_t/(1 + g_t), g_t)$

        – $v_t = (c_t/(1 + g_t), -1)$

        – $R_t = \begin{pmatrix} R_{t-1} & 0 \\ 0 & 0 \end{pmatrix}$

        – $R_t = \text{cholup}(R_t, u_t, '+')$

        – $R_t = \text{cholup}(R_t, v_t, '-')$

    – $\alpha_t = R_t^{-1} R_t^{-T} K_t^T (Y_t, 0)$

    – $b_t = (k(x_t, x_j))_{j \in \mathcal{I}_t}$

    – predict $\widehat{y}_t = b_t^T \alpha_t$

    – receive $y_t \in \mathbb{R}$

    – update $Y_t = (Y_{t-1}, y_t)$

## H.2 Python code

```python
import numpy as np
from math import factorial

class PhiAWV:
    def __init__(self, d, sigma=1.0, lbd=1.0, M=2):
        self.b = None
        self.A_inv = None
        self.M = M
        self.lbd = lbd
        self.sigma = sigma

    def taylor_phi(self, x):
        res = np.array([1.])
        mm = np.array([1.])
        for k in range(1,self.M+1):
            mm = (np.outer(mm, x)).flatten()
            q = self.sigma**k*np.sqrt(factorial(k))
            res = np.concatenate((res,mm/q))
        c   = 2*self.sigma**2
        res *= np.exp(-np.dot(x,x)/c)
        return np.array(res)

    def predict(self, x):
        z = self.taylor_phi(x)
        if self.b is None:
            r = len(z)
            self.b = np.zeros(r)
            self.A_inv = (1/self.lbd)*np.eye(r)

        v = np.dot(self.A_inv, z)
        v /= np.sqrt(1 + np.dot(z, v))
        self.A_inv -= np.outer(v, v)
        w_hat = np.dot(self.A_inv, self.b)
        self.z = z
        return np.dot(w_hat, z)

    def update(self, y):
        self.b += y*self.z

def update_inv(A_inv,x):
    B = x[:-1][:,None]
    C = np.transpose(B)
    D = np.array(x[-1])[None,None]
    if A_inv.size == 0:
        return 1./D
    compl = 1./(D - np.dot(np.dot(C,A_inv),B))
    R0 = A_inv + np.dot(np.dot(np.dot(np.dot(A_inv,B),compl),C),A_inv)
    R1 = - np.dot(np.dot(A_inv,B),compl)
    R2 = - np.dot(np.dot(compl,C),A_inv)
    R3 = np.array(compl)
    return np.block([[R0, R1], [R2, R3]])

def KORS(x, KMM, S, SKS_inv, lbd=1., eps=0.5, beta=1.):
    kS = KMM[:,-1]*S
    SKS = np.array(S)[None,:] * KMM * np.array(S)[:,None]
    en = np.eye(len(kS))[:,-1]
    SKS_inv_tmp = update_inv(SKS_inv, kS + lbd*en)
    tau = (1+eps)/lbd*(KMM[-1,-1] - np.dot(kS, np.dot(SKS_inv_tmp, kS)))
    p = max(min(beta*tau,1),0)
    z = np.random.binomial(1,p)
    S = S[:-1]
    if z:
        S.append(1/p)
```

```python
            SKS_inv = update_inv(SKS_inv, 1/p*KMM[:,-1]*S + lbd*en)
        return z, S, SKS_inv

class Nystrom_kernel_AWV:
    def __init__(self, d, k, lbd=1.):
        self.c = np.zeros(0)
        self.X = np.zeros((0,d))
        self.Y = np.zeros(0)
        self.k = k
        self.lbd = lbd
        self.R = np.eye(0)
        self.chosen_idx = []
        self.KnM = np.eye(0)
        self.S = []
        self.SKS_inv = np.eye(0)

    def predict(self, x):
        self.X = np.concatenate((self.X,x[None,:]), axis=0)
        n = self.X.shape[0]
        Kn = np.array([self.k(self.X[i,:],x) for i in self.chosen_idx])
        self.KnM = np.concatenate((self.KnM,Kn[None,:]), axis=0)
        K_kors = np.concatenate((self.KnM[self.chosen_idx+[n-1],:],
                                 np.concatenate((Kn, [self.k(x, x)]))[:,None]),
                                            axis=1)
        z, self.S, self.SKS_inv = KORS(x, K_kors, list(self.S)+[1], \
                                            self.SKS_inv, lbd=self.lbd)
        self.R = cholup(self.R, self.KnM[-1,:], '+')
        if z:
            self.chosen_idx.append(n-1)
            M = len(self.chosen_idx)
            KM = np.array([self.k(self.X[i,:],x) for i in range(n)])
            self.KnM = np.concatenate((self.KnM,KM[:,None]), axis=1)
            a = self.KnM[:,-1].T
            d = np.dot(a, a) + self.lbd*self.KnM[-1,-1]
            if M == 1:
                self.R = np.array([[np.sqrt(d)]])
            else:
                b = self.KnM[self.chosen_idx[:-1],-1]
                c = np.dot(self.KnM[:,:-1].T, a) + self.lbd*b
                g = np.sqrt(1 + d)
                u = np.concatenate((c/(1+g), [g]))
                v = np.concatenate((c/(1+g), [-1]))
                self.R = np.block([[self.R, np.zeros((M-1,1))],
                                   [np.zeros((1,M-1)), 0]])
                self.R = cholup(self.R, u, '+')
                self.R = cholup(self.R, v, '-')
        Yp = np.concatenate((self.Y, [0]))
        if len(self.R) > 0:
            self.c = solve_triangular(self.R,
                    solve_triangular(self.R.T, np.dot(self.KnM.T,self.Y),
                                    lower=True))
        Kn = np.array([self.k(self.X[i,:],x) for i in self.chosen_idx])
        return np.dot(Kn, self.c)

    def update(self, y):
        self.Y = np.concatenate((self.Y,np.array(y)[None]), axis=0)
```