[Reviews · NeurIPS 2019]

Reviewer 1



1. Originality: The algorithm and analysis in the paper is novel. 2. Quality: The method is supported by any essential theoretical analysis and experiments on some large scale regression datasets. 3. The overall clarity of the paper is good.

Reviewer 2



# Originality I am not an expert in the literature in online learning with kernels, but it seems [12, 13] are indeed the most relevant prior work, which the paper cites and builds upon. I would be surprised if the idea of using Taylor approximations for kernels is completely new; please consider adding relevant references. # Quality I have not verified the details of all the proofs, but the math in general does seem to add up: The main claims do seem to follow correctly from the central Theorem 9 and the other relevant results. So does Theorem 3, which is central to the third contribution which I find to be the most interesting. I believe, however, that the claims in Section 3.1. regarding Gaussian kernels have to be taken with caution. In particular, the computational and storage complexity does not scale with the effective dimension as claimed, apparently, but only with a quantity that is an upper-bound on the effective dimension. As such, it cannot be claimed that the algorithm is more efficient than the $O(n^2)$ ones (and it is not a really useful property to be more efficient that $O(n^2)$ whenever $n=O(e^d)$). Having said that, the authors do provide experimental results to show that the algorithm with taylor approximation is fast. # Clarity The paper is clearly written and easy to follow, though it certainly requires a proof-reading as there are many typos, gramatical errors, etc. The proofs also seems cleanly written and easy to follow. # Significance I have commented on the significance in the previous question. In general, this is a clear paper and a nice addition to the literature, hence I am recommending acceptance. However, I have some questions about the third contribution (see below), and the first and second contributions do not seem enough on their own. # Questions for the authors: - Is there any other benefit to your use of AWV other than removing the bound on $||f||$? Wouldn't Pros-N-KONS obtain the same regret bound as PKAWV if you apply Theorem 9 (with the difference that the first two terms ("regret terms") in the bound are handled by the KONS regret analysis rather than the K-AWV analysis)? In particular, is a version of Corollary 8 possible for KONS? - What is the key property that makes Corollary 8 possible? I realize that the approximation terms in Theorem 9 capture the change in the projection operators, but shouldn't this be compensated for by the adaptivity of the projections? Also, why is the $m$ term not there in the projection error term bound of Pros-N-KONS (why is it only in the regret bound)? ####################################### After the author response: ####################################### After reading the other reviews as well as the author response, I would like to increase my score. The author response does answer my questions for the most part. I can also agree with the authors that there are $(n,d)$ regimes in which this is applicable and useful, and the authors also correctly note that the result is more general, and less computation is possible at the expense of more regret.

Reviewer 3



While the paper makes some good theoretical contributions, I do feel that the contributions may be rather niche, and of limited impact. Moreover, the exposition was hard to follow. Several undefined notations and grammar mistakes make the reading unpleasant. Originality: I would rate the ideas low on originality, as the idea of using Taylor expansion and Nystrom approximation has been extensively explored for online learning with kernels (and in general learning with kernels). Quality and Clarity: - There are several grammatical and semantic/notation errors which disrupt a smooth reading (e.g. we aims” at; little “considers” non-parametric function; AWV seems like an arbitrary term for denoting an algorithm – and I could not find what AWV stands for; What is B and d_eff in Eq 2?; Line 122 – is not “The on the space…”; and many more such instances throughout the paper ) - First paragraph in the introduction makes several strong claims about the research directions in general - not properly validated through references or other evidence - What are the pros and cons of considering squared loss over a Lipschitz loss? - What was the motivation of choosing the specific basis function for the Gaussian kernel? How does this approach differ in principle from other projection methods (e.g. Fourier projection) - Since the dictionary for Nystrom method is fixed, would that make the method sensitive to noise? - The experiments show algorithm as PKRR, while the algorithm has been denoted as PKAWV in most of the paper. Experiment section stats that M = 2,3,4 are tried, but only results for 2 and 3 are shown - I am not sure why the proposed approaches are faster than FOGD, as all of them are essentially linear projection algorithms. All algorithms should grow linearly in time (except KRR), but from experiments do not seem to indicate this for the baselines. Time out of 5 minutes (and later stated as 10 minutes) seems to be very small and arbitrary. Some of the learning curves also seem to be a bit strange. How significant is the performance gain of the proposed methods? The log scale graphs make it hard to see – perhaps a table with standard deviations gives a better picture. - Were the classification experiments done using a squared loss too? Would that make the comparison slightly unfair to favour the proposed method? Significance: While the work in general is good from a pure theoretical perspective, I think that, the audience for this work might be a bit too narrow (focusing on one specific type of loss function, and primarily improving the speed and maintaining the same regret bounds), limiting its significance.

Reviewer 4



Originality/related work: The paper feels incremental as it combines existing techniques/ideas in a (perhaps) novel fashion (perhaps the main novelty is bringing in the Vovk-Azoury-Warmuth forecaster). The main claim is that this is the first method that achieves subquadratic per-round complexity (for reasonable classes of kernels). In relation to this, it would be great if the authors could comment on the relation of the present work and the papers [1,2,3]. In particular, the recent paper by Zhang and Liao seems to be doing something very close to the present paper, though they consider general convex losses (unlike the present paper which considers squared loss), hence their regret results will be different. However, the essence of both papers seem to be the same (this paper is also close to [16], cited in the paper). Calandriello et al's COLT paper [2] considers optimization with bandit feedback, but otherwise the ideas are very close. The same applies to [3], which I would have also expected to be cited and compared with. Quality/clarity: The paper is generally well written. It is a bit hard to make out the algorithms though based on the text of the main paper. Some minor comments are listed below. Significance: The method looks promising; the main theoretical contribution is perhaps Theorem 3 (or, its more advanced version, Theorem 9). Minor comments and some questions: * abstract mentions adversarial datasets; this seems like a nonstandard terminology that is perhaps not needed * e^d seems large.. * Throughout the paper, paper numbers are used with no names. This makes the paper unnecessarily hard to read. Please write citations in a way that allows mortal humans to follow them. * After Eq (2) it is mentioned that the bound in Eq (2) is "essentially optimal" and the next section will explain this; but I could not really find where this is explained. Please be more precise with cross-references. I guess this refers to minimax lower bounds by Zhang-Duchi-Wainwright; I'd appreciate if the appendix had the precise statement (reformulated in the formalism of this paper). * Page 2, line 50: aims->aim * Page 3, line 88: player->learner * line 99: what about larger gamma?? * line 109: "rough inequality" -> "approximate inequality"? Comma missing after Finally. * line 119: f = sum alpha_i phi(x_i) is not formally correct * line 120: K_pp has not been introduced yet * page 3, line 122: First sentence is weird. * page 4, line 167: C is the covariance operator: which one? * line 168: Euclidean projection wrt the RKHS norm, right? * page 6, line 202: So what is the range of the grid? Can you add more details? citing the CBL book does not help much here * line 209: "works"->"work". What does "It" refer to? Consider rephrasing. * line 213: So where is that algorithm? * line 235: classical assumption: perhaps "common setting"? (this is not really an assumption; more like a setting or an example) * page 8, Fig 2 (also in appendix): Figures seem to use labels that don't match the main body. References --------------- [1] Zhang, X., & Liao, S. (2019, May). Incremental Randomized Sketching for Online Kernel Learning. In International Conference on Machine Learning (pp. 7394-7403). [2] Calandriello, D., Carratino, L., Lazaric, A., Valko, M., & Rosasco, L. (2019). Gaussian process optimization with adaptive sketching: Scalable and no regret. arXiv preprint arXiv:1903.05594. (also at COLT 2019) [3] Rudi, A., Calandriello, D., Carratino, L., & Rosasco, L. (2018). On fast leverage score sampling and optimal learning. In Advances in Neural Information Processing Systems (pp. 5672-5682).

[Author Response · NeurIPS 2019]

We warmly thank the reviewers for their careful reading and their remarks.

**General comments** We would like to point out that while Taylor's expansion or Nyström have already been explored for i.i.d. data, the existing results are limited in the adversarial context. No algorithm reached optimal regret with a per-round computational complexity $o(n^2)$. In particular, the existing analysis of Nyström (for Pros-N-Kons [13] or FOGD [16]) must restart the algorithm to update the approximation. This leads to deterioration of the regret bounds and poor practical performance. Removing these restarts was not trivial but crucial to achieving optimal rates.

We agree that the algorithms and approximation techniques used in the paper are not new (Kernel Ridge, Nyström, Taylor's approximation). Yet, surprisingly they outperform on the 1) computational complexity, 2) theoretical guarantees (regret) and 3) practical performance (see experiments) more complex algorithms (using restarts, additional projections,...) that are widely used in practice and constitute the actual state of the art. We think it is is interesting in itself to show new optimal theoretical guarantees for adversarial variants of these well-known methods.

Finally, we will take all the time necessary to carefully improve the writing and add appropriate references if necessary.

**To reviewer 4**

– *When $n^2 = O(e^d)$ none of the algorithms will be runnable.* We believe this is runnable in many real-life scenarios with a small number of explanatory variables but very long time series or large datasets. In online algorithms, a (close to) constant cost per iteration is of great importance. For example, when $d = 10$, we have $n^2 \gg e^d$ as soon as $n \gg 150$ which is likely to happen. In this article, we propose compelling experiments on large-scale datasets from many contexts that adapt to our context. In addition, as shown in Figure 1 or Figure 2 with $M = 2$, there is an approximation trade-off: less complexity is always possible at the cost of a sub-optimal regret.

– *Similar results for Pros-N-Kons and interest of AWV.* We guess (though it was not proved) that Pros-N-Kons may achieve similar results to Cor. 8. But this is not as clear for Thm. 9 which allows optimal rates to be obtained for many Kernels when the features are revealed sequentially. The analysis of changing approximations (without restarting the algorithm) was not trivial. It was quite specific to squared loss and KAWV. As shown in Fig. 1 (left), due to restarts, the performance of Pros-N-Kons is capped. Its current analysis does not allow it to get optimal regret.

– *Why is the $m$ term not there in the projection error term bound of Pros-N-Kons?* The analysis of Pros-N-Kons is different and can only deal with fixed approximations and the algorithm needs to be restarted $m$ times. Hence a regret bound deteriorated by a factor $m$. Our algorithm is not restarted but the approximation term is directly controlled but get multiplied by $m$. It yields a tighter upper-bound on the regret. Besides, note that $\mu$ and $m$ are linked since $m \approx d_{\text{eff}}(\mu) \leq (n/\mu)^\gamma$. The approximation term of Pros-N-Kons therefore also depends on $m$.

**To reviewer 5** First, we will have our submission proof-read for English style and grammar issues. Thanks for the typos, we will check the paper carefully.

– *Narrow audience (restricted to squared loss, improved speed but same regret bounds)* We believe that our work may be of interest to a wide audience because: 1) squared loss is widely used 2) existing algorithms that achieve the same (optimal) regret suffer from a total time complexity of order $n^3$ that is prohibitive for many applications.

– *Squared loss vs Lipschitz loss and comparison to [16].* Much faster rates are possible for the squared loss. In particular, the regret bound provided by [16] for Lipschitz losses (see Th. 1) is much worse than ours because:
  - it depends on the $\ell_1$-dual norm which can be arbitrarily large for ill-conditioned data while ours only depends on the $\ell_2$-norm and the effective dimension;
  - it is of order $\sqrt{n}$ while ours can be logarithmic.

– *Fixed Nyström dictionary and noise sensitivity.* The dictionary for Nyström is not fixed but growing randomly as new samples are observed. We consider an adversarial setting, the algorithm should therefore be robust to noise.

– *Why are our approaches faster than FOGD?* FOGD is actually the fastest algorithm. However, it has worse theoretical regret bound and poor performance in our experiments.

– *Classification experiments with squared loss.* The squared loss was used for learning but the 0/1 loss was used to evaluate the algorithms. We re-used the experimental setups of previous work [13,16] to ease the comparison and reproducibility. It should be noted that Pros-n-Kons makes a *curvature* assumption on the loss that prevents the use of losses such as hinge or logistic (possible but at the price of a exponentially small hyperparameter). So it is also natural to use Pros-n-Kons with squared loss. For the sake of coherence we used FOGD with squared loss too.

– *The experiments use PKRR rather than PKAWV.* Our experiments on real datasets are closer to an i.i.d. setting rather than an adversarial one. In this case, Kernel Ridge Regression (KRR) seems to be more suited than PKAWV (its adversarial counterpart). We will report the results of PKAWV which are very similar to PKRR. Furthermore, we can run the experiments longer. But since the results are reported in a loglog scale and since the rates are already visible, this will not add much information.

[Meta-Review · NeurIPS 2019]

The paper presents solid theoretical contributions for online kernel regression, illustrated with experiments, While the novelty of the work is somewhat limited, the results seem interesting and non-trivial. Thus, based on the reviews (a new expert review was solicited to increase the confidence in the reviewers' pool) and my own reading, I am happy to recommend acceptance.